# Learning Memory-Enhanced Improvement Heuristics for Flexible Job Shop Scheduling

**Jiaqi Wang[1,2], Zhiguang Cao[4], Peng Zhao[1,3]∗, Rui Cao[1,3],**
**Yubin Xiao[1,3], Yuan Jiang[5]∗, You Zhou[1,2,3]∗**

[1]Key Laboratory of Symbolic Computation and Knowledge
Engineering of Ministry of Education, Jilin University
[2]College of Software, Jilin University
[3]College of Computer Science and Technology, Jilin University
[4]Singapore Management University  [5]Nanyang Technological University

{jqwang24, pengzhao23, ruicao24, xiaoyb21}@mails.jlu.edu.cn
zhiguangcao@outlook.com, yuan005@e.ntu.edu.sg, zyou@jlu.edu.cn

## Abstract

The rise of smart manufacturing under Industry 4.0 introduces mass customization and dynamic production, demanding more advanced and flexible scheduling techniques. The flexible job-shop scheduling problem (FJSP) has attracted significant attention due to its complex constraints and strong alignment with real-world production scenarios. Current deep reinforcement learning (DRL)-based approaches to FJSP predominantly employ constructive methods. While effective, they often fall short of reaching (near-)optimal solutions. In contrast, improvement-based methods iteratively explore the neighborhood of initial solutions and are more effective in approaching optimality. However, the flexible machine allocation in FJSP poses significant challenges to the application of this framework, including accurate state representation, effective policy learning, and efficient search strategies. To address these challenges, this paper proposes a **M**emory-enhanced **I**mprovement **S**earch framework with he**t**erogeneous gr**a**ph **r**epresentation—*MIStar*. It employs a novel heterogeneous disjunctive graph that explicitly models the operation sequences on machines to accurately represent scheduling solutions. Moreover, a memory-enhanced heterogeneous graph neural network (MHGNN) is designed for feature extraction, leveraging historical trajectories to enhance the decision-making capability of the policy network. Finally, a parallel greedy search strategy is adopted to explore the solution space, enabling superior solutions with fewer iterations. Extensive experiments on synthetic data and public benchmarks demonstrate that *MIStar* significantly outperforms both traditional handcrafted improvement heuristics and state-of-the-art DRL-based constructive methods.

## 1 Introduction

As an emerging paradigm integrating advanced manufacturing and digital technologies, Industry 4.0 is transforming production toward dynamic, large-scale, and customized manufacturing [1, 2]. The job-shop scheduling problem (JSP) [3], a classic NP-hard combinatorial optimization problem (COP), has been widely studied and plays a critical role in manufacturing systems [4]. As an extension of JSP [5, 6], the flexible job-shop scheduling problem (FJSP) allows each operation to be assigned to

---

∗Corresponding author.

39th Conference on Neural Information Processing Systems (NeurIPS 2025).

one of several eligible machines [7], making it more suitable in handling the flexibility and diversity of task–resource relations in new manufacturing paradigms [8]. While this increased flexibility enables FJSP to better meet the demands of mass customization and automation in smart manufacturing (SM) [5, 9, 10], it also introduces greater challenges in developing effective solution strategies [11].

Recent advances in neural models have shown remarkable effectiveness in solving complex COPs [12, 13, 14, 15, 16, 17, 18]. As one promising paradigm among them, Deep Reinforcement Learning (DRL) formulates the scheduling task as a Markov decision process (MDP), learning a parameterized policy network that receives scheduling state as input and outputs feasible actions. DRL approaches to scheduling can be categorized into *construction* and *improvement* [19, 20]. *Construction* methods build schedules by assigning an operation to a machine at each step, progressively extending partial solutions to complete ones [21]. However, their performance heavily relies on state representations [22]. Many studies [23, 24, 25, 26] model scheduling states using disjunctive graphs, but the incompleteness of partial solutions often leads to the omission of important components [21] (e.g., the disjunctive arcs among undispatched operations [23]). Additionally, disjunctive graphs struggle to incorporate essential work-in-progress information required during construction [21] (e.g., current machine load and job status). These limitations lead to suboptimal schedules and reduce their adaptability in SM. In contrast, *improvement* methods start from a complete initial solution and iteratively refine it through small adjustments [27]. As the MDP state represents a fully scheduled solution, it avoids the information loss inherent in partial ones. Moreover, the structure of disjunctive graphs naturally encodes topological relationships among operations [28], enabling complete schedules to be effectively represented with all necessary information [21]. This better supports the high-performance demands of complex scheduling scenarios in SM.

Despite the promising results, most existing DRL-based improvement methods focus only on nonflexible problems, such as the JSP [21, 29]. The increased complexity of FJSP over JSP poses significant challenges for designing effective improvement approaches. First, the one-to-many relationships between operations and machines make the scheduling state more intricate [8], challenging its representation and demanding more expressive neural networks for encoding. Second, improvement methods explore solutions through local moves within neighborhoods. In FJSP, due to the complex decisions involving both operation sequencing and machine assignment [6], this process requires tailored local moves as actions in the MDP [27]. Moreover, the existence of flexibility in FJSP significantly enlarges the solution space, increasing the risk of local optima entrapment and necessitating more efficient search strategies to reduce the extensive number of iterations for convergence [27, 30].

To address the above challenges, we propose a DRL-based improvement heuristic framework for FJSP. First, to represent complex scheduling states, we design a novel heterogeneous disjunctive graph by adding machine nodes with directed hyper-edges to encode the complete solution, effectively capturing critical information about operation sequencing on machines. We also propose a heterogeneous graph neural network and enhance its exploratory capability via a memory module, which stores compact representations of visited solutions and retrieves relevant information to enrich the current state embedding. This module leverages historical schedules to improve decision-making and alleviate local optima issues. Second, we construct the action space based on the Nopt2 neighborhood structure, enabling simultaneous adjustment to operation sequences and machine assignments, and further reduce its dimensionality via constraint relationships to enhance search efficiency. Finally, we propose a parallel greedy exploration strategy that evaluates multiple candidate actions at each step, achieving comparable solution quality with fewer iterations. Extensive experiments on synthetic data and public benchmarks demonstrate the superior performance of our approach in solving FJSP.

The contributions of this paper are summarized as follows: 1) The first DRL-based improvement heuristic framework for FJSP—*MIStar*, capable of learning size-agnostic policies that outperform both traditional handcrafted improvement and state-of-the-art DRL construction methods. 2) An MDP formulation with a heterogeneous disjunctive graph for state representation of complete scheduling solutions in FJSP, and an action space that enables simultaneous adjustments to operation sequences and machine assignments. 3) A memory-enhanced heterogeneous graph neural network (MHGNN) that leverages historical solutions to improve decision-making and alleviate local optima issues. 4) A parallel greedy exploration strategy that improves efficiency of solution space exploration.

## 2 Related work

This section reviews the limitations of DRL-based construction methods and recent advances in improvement methods, highlighting the challenges in solving FJSP.

*Construction* methods construct complete solutions by assigning an operation to a machine at each step. Compared to the heuristic guidance offered by conventional dispatching rules, DRL employs DNNs to score actions, but their effectiveness heavily relies on state representations [22, 31, 32]. Early approaches [33, 34, 35, 36] extract multiple general state features as input, which overcompresses state information and neglects the structural nature of FJSP [37]. Recent works [8, 23, 24, 38, 39] integrate GNNs with DRL by modeling scheduling states as graphs, learning rich graph embeddings with structural information for decision-making. However, the incompleteness of partial solutions often results in the omission of important components [21] (e.g., the disjunctive arcs among undispatched operations [23]), and work-in-progress information required during construction is hard to explicitly represent in disjunctive graphs [21], leading to suboptimal performance.

*Improvement* methods iteratively refine solutions by performing local moves within the neighborhood. Conventional methods use handcrafted rules to greedily select the best solution at each step, but suffer from myopia and costly neighborhood evaluation [21]. Recent studies [21, 29, 40, 41] have addressed these limitations by leveraging deep policy networks in DRL to directly generate local moves. Such DRL-based improvement heuristics were initially applied to routing problems [42, 43, 44] and have since been adapted to scheduling domains. By representing complete solutions as MDP states through disjunctive graphs, these approaches circumvent the inaccuracies in state representation that arise from partial solutions [21]. Falkner et al. [40] train a GNN-based JSP solver using DRL to adapt local search strategies according to problem-specific features. However, their design of local operators relies heavily on expertise, limiting the generalizability. Closer to our work, Zhang et al. [21, 29] represent JSP solutions with disjunctive graphs and define the action space as a set of exchangeable operation pairs based on the N5 neighborhood structure [45], training a GNN-based policy network via DRL. While effective for JSP, extending this framework to FJSP—whose flexible machine assignments better suit mass customization and dynamic production in Industry 4.0 [9, 10]—faces critical challenges. First, conventional disjunctive graphs lack machine nodes, making them insufficient for complex states of FJSP. In addition, the N5-based action design does not account for machine assignment adjustments(See Appendix A for analysis). Furthermore, the increased flexibility of FJSP over JSP enlarges the solution space and aggravates local optima issues in local search [46].

An effective approach to this issue is leveraging memory mechanisms by incorporating past experience to promote exploration [47]. Garmendia et al. [48] maintain a tabu memory to filter redundant visits in routing problems, yet such methods fail to integrate historical data into the decision-making of the policy, leaving historical information underutilized. A closely related work is MARCO [47], which aggregates contextually relevant information from the memory module to enhance state embeddings and improve policy decisions. However, its application in improvement methods is limited to simple binary optimization problems (i.e., maximum cut and maximum independent set), and its approach to information aggregation is unsuitable for the complex constraints in FJSP. In contrast, we simplify scheduling representations, which are stored in the memory module and aggregated via a soft voting mechanism [49] to better accommodate the intricate constraints of FJSP.

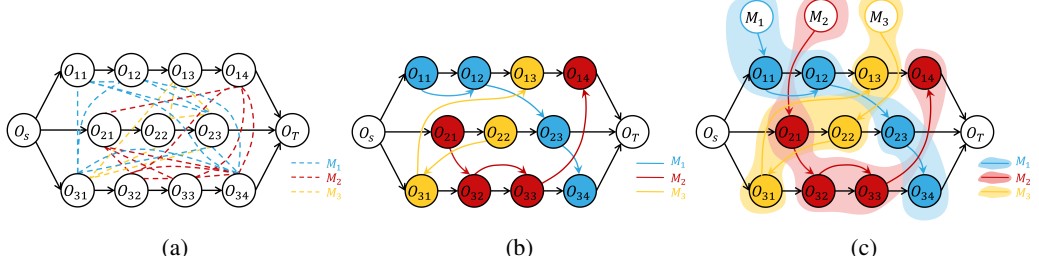

Figure 1 **Graph representations of FJSP.** (a) FJSP instance in a disjunctive graph; (b) feasible solution in a disjunctive graph; (c) feasible solution in our graph.

## 3 Preliminaries

This section presents the formulation of FJSP, disjunctive graphs, and Nopt2 neighborhood structure.

**Flexible job shop scheduling problem.** FJSP is defined as follows: Given a set of jobs $\mathcal{J} = \{J_1, J_2, \cdots, J_n\}$ and a set of machines $\mathcal{M} = \{M_1, M_2, \cdots, M_m\}$, each job $J_i$ consists of $n_i$ operations denoted by $\mathcal{O}_i = \{O_{i_1}, O_{i_2}, \cdots, O_{in_i}\}$. The operations within a job must be processed in a specific order, known as precedence constraints. Each operation $O_{ij}$ can be processed by any machine $M_k$ for a processing time $p_{ij}^k$ from its compatible set $\mathcal{M}_{ij} \subseteq \mathcal{M}$, and the processing is non-preemptive. Each machine can process only one operation at a time. The objective of FJSP is to assign each operation to a compatible machine and determine its processing sequence on the selected machine to minimize the makespan, defined as the maximum completion time $C_{\max} = \max_{i,j}\{C_{ij}\}$, where $C_{ij}$ denotes the completion time of $O_{ij}$.

**Disjunctive graph.** An FJSP instance can be represented by a disjunctive graph $G = (\mathcal{O}, \mathcal{C}, \mathcal{D})$ [50]. The operation set $\mathcal{O} = \{O_{ij} \mid \forall i, j\} \cup \{O_S, O_T\}$ includes all operation nodes and two dummy ones representing the start and end states. The conjunctive arc set $\mathcal{C}$ consists of directed arcs that describe the precedence constraints between operations of the same job. The disjunctive arc set $\mathcal{D} = \cup_k \mathcal{D}_k$ comprises undirected edges, and each $\mathcal{D}_k$ connects operations processed on the same machine $M_k$. In FJSP, each operation can be connected to multiple disjunctive arcs. Therefore, solving an FJSP instance is equivalent to selecting a disjunctive arc for each node and fixing its direction to construct a DAG [28]. In the graph, the longest path from $O_S$ to $O_T$ is called the critical path (CP), whose length represents the makespan of the solution. See Figure 1a and 1b for examples.

**The Nopt2 neighbourhood Structure.** Given a solution $s$, the Nopt2 constructs the neighbourhood $Nopt2(s)$ as follows. First, the critical path CP(s) is identified. Then, an operation $O_{ij}^k$ on the critical path is chosen and deleted from its current processing sequence of $M_k$. It is reinserted into the optimal insertion interval within the sequence of an alternative compatible machine $M_{k'}$ yielding a new solution $s'$. The optimal insertion interval is determined based on the precedence relations between the newly

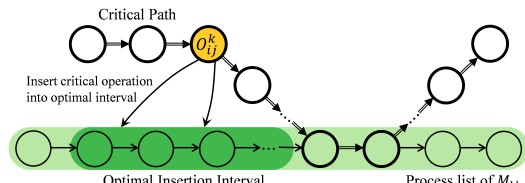

Figure 2 **Nopt2 neighborhood structure.**

inserted operation and existing operations on $M_{k'}$, ensuring that all constraints are satisfied and the makespan of $s'$ is potentially reduced. We adopt the Nopt2 neighborhood structure [51] (Figure 2) to define the local move during the search process of *MIStar*. By identifying optimal insertion interval, this structure significantly reduces the size of neighborhood. The neighborhood size $|Nopt2(s)|$ is influenced by both the instance scale and the flexibility of machine selection.

## 4 Methodology

This section details our framework, illustrated in Figure 3. Given an FJSP instance, an initial solution is generated and transformed into a heterogeneous graph, from which a memory-enhanced GNN learns embeddings. The policy network samples several local moves from the neighborhood and evaluates in parallel. The best one is executed to update the current solution, yielding a new one. The process terminates after a predefined search horizon. We formulate the iterative solution optimization as a MDP. This section first formalizes the MDP, then presents a novel heterogeneous graph representation and the memory-enhanced GNN-based policy network. Finally, we describe the parallel greedy exploration strategy and the training algorithm.

### 4.1 MDP Formulation

We formulate the iterative optimization process of the FJSP as a discrete MDP. At each step $t$, the agent, the decision-maker based on a given policy, selects an action (local move) from the action space (neighborhood), and the state transitions accordingly to represent a new solution. The MDP is defined as follows.

**State.** The state $s_t$ encodes a complete solution and is defined by a set of feature vectors, including $\mathbf{x}_{o_{ij}} \in \mathbb{R}^9$ for each operation $O_{ij}$ and $\mathbf{y}_{M_k} \in \mathbb{R}^4$ for each machine $M_k$, detailed in Appendix B.
**Action.** The action space $\mathcal{A}_t$ comprises local moves in the Nopt2 neighborhood, changing with

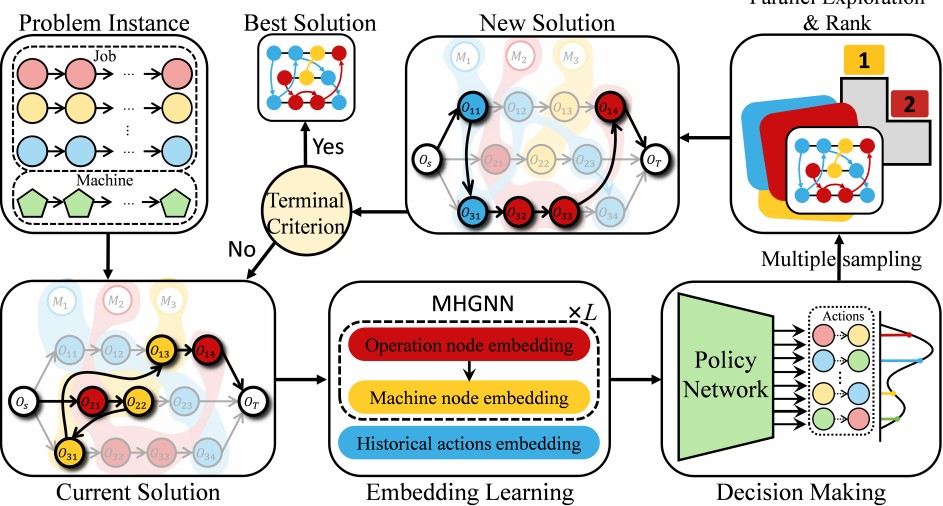

Figure 3 **The architecture of *MIStar*.**

the state $s_t$. Each action $a_t = [O_m, O_n, M_k]$ denotes inserting critical operation $O_m$ before operation $O_n$ in the processing sequence of machine $M_k$, where $O_m \notin \{O_S, O_T\}$ and $O_n \neq O_S$.
**Reward.** The reward comprises two components: improvement in solution quality $r_{gain}$ and penalty for redundant solution visits $r_{penalty}$. The term $r_{gain}$ measures the improvement over the incumbent best solution $s_t^*$ (initialized as $s_0$), and is defined as: $r_{gain} = \max(C_{\max}(s_t^*) - C_{\max}(s_{t+1}), 0)$, where maximizing the cumulative reward is equivalent to minimizing the makespan $C_{\max}(s_t^*)$. The term $r_{penalty}$ discourages redundant exploration by evaluating similarity between $s_t$ and previous states. Formally, it is computed as: $r_{penalty} = \lambda \times \frac{1}{K} \sum_{\omega \in \Omega_{\text{TopK}}} \omega$, where $\Omega_{\text{TopK}}$ denotes the top-$K$ similarity scores between $s_t$ and stored states, and $\lambda$ is the penalty coefficient. The overall reward is defined as $r_t(s_t, a_t) = r_{gain} - r_{penalty}$, which is initially dominated by makespan improvement and gradually shifts toward solution diversity as improvements diminish.

## 4.2 Directed Heterogeneous Graph

Conventional disjunctive graphs lack machine nodes, hindering GNNs from accurately capturing machine states in FJSP, which in turn limits policies to effectively maximize machine utilization [37]. Inspired by the heterogeneous disjunctive graph $\mathcal{H}$ [8], we introduce a novel directed heterogeneous disjunctive graph $\overrightarrow{\mathcal{H}} = (\mathcal{O}, \mathcal{M}, \mathcal{C}, \mathcal{E})$ to represent scheduling solutions. In our graph, the operation and machine nodes are denoted as $\mathcal{O}$ and $\mathcal{M}$, while the conjunctive arc set $\mathcal{C}$ and hyper-edge set $\mathcal{E}$ represent the processing sequences on jobs and machines, respectively. Mathematically, the hyper-edge is a special edge that can join any number of vertices [52]. Since machine assignment and processing sequence are fixed in FJSP solutions, we reinterpret $\mathcal{E}$ as a directed hyper-arc set of size $|\mathcal{M}|$, i.e., $\mathcal{E} = \{E^k = (M_k, O^{k1}, O^{k2}, \ldots, O^{kn_k})\}$, where $n_k$ is the number of operations processed on $M_k$. As shown in Figure 1c, each machine nodes $M_k$ connects to its processing sequence via directed hyper-arcs $E^k$ to explicitly encode machine status, and the unified directed arcs in $\overrightarrow{\mathcal{H}}$ enable clearer delineation and distinction of different solutions. See Appendix C for analysis.

## 4.3 Memory-enhanced Heterogeneous Graph Neural Network

We propose a memory-enhanced heterogeneous graph neural network (MHGNN) to extract state features from scheduling solutions. As shown in Figure 4, MHGNN encodes the topological and sequential constraint information to generate operation embeddings, which are aggregated into machine nodes to obtain machine embeddings. Meanwhile, relevant historical information is retrieved and extracted as historical action embeddings, which are then concatenated with machine and operation embeddings before being fed into the policy network to produce an action distribution. The following sections detail the embeddings design for operations, machines, and historical information.

### 4.3.1 Operation Node Embedding

Following [21], we focus on two key aspects of the FJSP graph: (1) the topology dynamically changes due to the transitions of MDP state, and (2) rich semantics are conveyed through two types of neighbors—job predecessors and machine predecessors—reflecting precedence constraints and processing order on machines via conjunctive arcs and hyper-arcs, respectively. Their joint representation is crucial for distinguishing solutions, forming the foundation for scheduling decisions.

To encode the topological information, we employ the Graph Isomorphism Network (GIN) [53], which is known for its strong discriminative power for non-isomorphic graphs. Specifically, given a graph $\overrightarrow{\mathcal{H}} = (\mathcal{O}, \mathcal{M}, \mathcal{C}, \mathcal{E})$, each operation $O_{ij} \in \mathcal{O}$ is encoded into a $q$-dimensional embedding through an $L$-layer GIN, where the $l$-th layer computes as follows:

$$\mu^l_{O_{ij}} = \text{MLP}^l \left( (1 + \epsilon^l) \cdot \mu^{l-1}_{O_{ij}} + \sum_{U \in N(O_{ij})} \mu^{l-1}_U \right) \tag{1}$$

where $\mu^l_{O_{ij}} \in \mathbb{R}^q$ denotes the topological embedding of operation $O_{ij}$ at layer $l$, initialized with its raw features $\mathbf{x}_{o_{ij}}$, $\epsilon^l$ is a learnable parameter, and $N(O_{ij})$ comprises predecessor operation nodes.

To capture rich semantic relationships among operations, we apply an $L$-layer Graph Attention Network (GAT) with $n_H$ attention heads, with the feature transformation at layer $l$ computed as:

$$\tau^l_{O_{ij}} = \text{GAT}^l \left( \tau^{l-1}_{O_{ij}}, \{\tau^{l-1}_U | U \in N(O_{ij})\} \right) \tag{2}$$

where $\tau^l_{O_{ij}} \in \mathbb{R}^q$ denotes the semantic embedding of operation $O_{ij}$ at layer $l$, initialized with raw features $\mathbf{x}_{o_{ij}}$. We obtain the operation embedding $h^l_{O_{ij}} \in \mathbb{R}^{2q}$ at layer $l$ by concatenating the above two embeddingshe final-layer outputs form the set of operation node embeddings $\{h_{O_{ij}} = h^L_{O_{ij}} \mid \forall O_{ij} \in \mathcal{O}\}$.

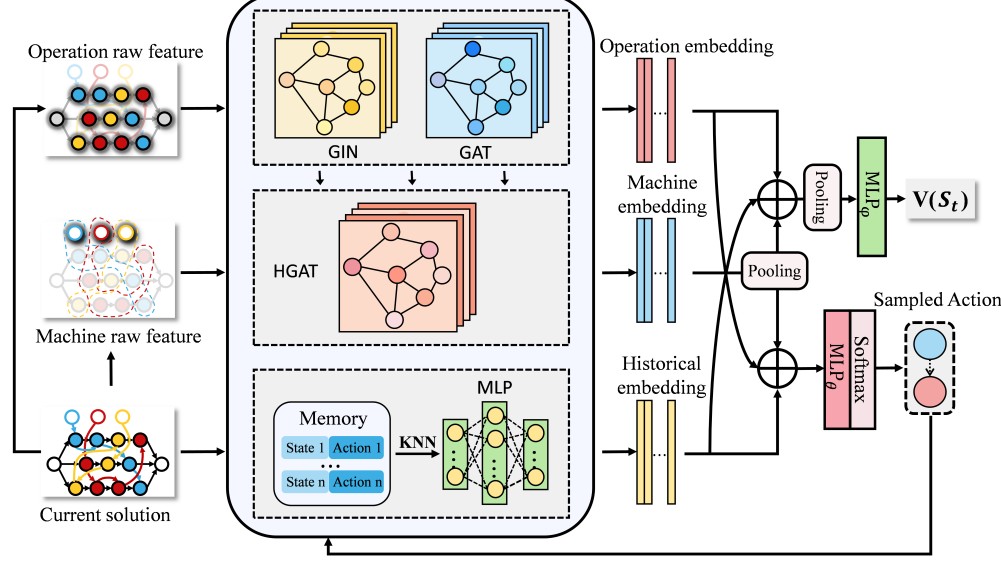

Figure 4 **The network architecture of the memory-enhanceou heterogeneou GNN.**

### 4.3.2 Machine Node Embedding

Machine embeddings are incorporated to capture machine load by aggregating the information of their associated operation nodes, serving as an essential heuristic to guide the policy in reassigning operations from high-load machines to low-load counterparts [8]. For each machine $M_k$, the hyper-arc $E^k = (M_k, O^{k1}, O^{k2}, \ldots, O^{kn_k})$ encodes its full processing sequence. Since the importance of operations at different positions varies, we employ an $L$-layer heterogeneous GAT [8] to extract machine node embeddings, where nodes with different attributes are processed by distinct linear

transformations rather than a shared one when computing the attention coefficients. The machine node embedding at layer $l$ is computed as follows:

$$h_{M_k}^l = \text{HGAT}^l \left( h_{M_k}^{l-1}, \{h_U^{l-1} | U \in N(M_k)\} \right) \tag{3}$$

where $h_{M_k}^l \in \mathbb{R}^q$ denotes machine embedding at layer $l$, initialized with raw features $\mathbf{y}_{M_k}$, $\text{HGAT}^l$ is a heterogeneous graph attention layer, and $N(M_k)$ comprises operations in its processing sequence. Similarly, the graph-level machine embedding is computed as $h_M = \frac{1}{|\mathcal{M}|} \sum_{M_k \in \mathcal{M}} h_{M_k}^L$.

### 4.3.3 Historical Action Embedding

At step $t$, we store the state-action pair $(s_t, a_t)$ in a memory module. To reduce storage and computational costs, we simplify the representation of $s_t$ by omitting node attributes and focusing solely on the processing sequence across machines (see analysis in Appendix D). Specifically, we store the operation-machine incidence matrix $\boldsymbol{L}_t$ at each step, where the non-zero element $(\boldsymbol{L}_t)_{i,j}$ represents the processing order index of operation $i$ on machine $j$, and assess the similarity between the current state $\boldsymbol{L}_t$ and each historical state $\boldsymbol{L}_{t'}$ ($t' < t$), by computing the Frobenius inner product as their similarity score $\omega_{t,t'} \in \Omega$, given by:

$$\omega_{t,t'} = \langle L_t, L_{t'} \rangle_F = \sum_{i=1}^{|\mathcal{O}|} \sum_{j=1}^{|\mathcal{M}|} (L_t)_{i,j} \cdot (L_{t'})_{i,j} \tag{4}$$

The $k$-nearest neighbors algorithm retrieves $K$ actions with the highest similarity scores, forming the set $\mathcal{A}_{L_t} = \{a_1, a_2, \ldots, a_K\}$. We aim to aggregate the information in $\mathcal{A}_{L_t}$ to enhance the state embedding and encourage exploration of more diverse solutions through the design of reward. However, each historical action $a_i = [O_{m_i}, O_{n_i}, M_{k_i}]$ is denoted by discrete integer indices indicating operations and machines. Simply weighted averaging yields semantically invalid continuous values, failing to provide useful guidance. To address this, we adopt a soft voting mechanism that aggregates the three dimensions separately based on frequency-weighted similarity, as follows:

$$\tilde{O}_m = \underset{O_m \in \mathcal{A}_{L_t}^1}{\arg\max} \sum_{i=1}^K \omega_{t,i} \cdot \mathbb{I}(O_{m_i} = O_m) \tag{5}$$

$$\tilde{O}_n = \underset{O_n \in \mathcal{A}_{L_t}^2}{\arg\max} \sum_{i=1}^K \omega_{t,i} \cdot \mathbb{I}(O_{n_i} = O_n) \tag{6}$$

$$\tilde{M}_k = \underset{M_k \in \mathcal{A}_{L_t}^3}{\arg\max} \sum_{i=1}^K \omega_{t,i} \cdot \mathbb{I}(M_{k_i} = M_k) \tag{7}$$

where $\omega_{t,i}$ is the similarity score between $s_t$ and the $i$-th most similar historical state, serving as the weight of $a_i$, $\mathbb{I}(\cdot)$ is the indicator function, and $\mathcal{A}_{L_t}^1$, $\mathcal{A}_{L_t}^2$ and $\mathcal{A}_{L_t}^3$ denote the unique value sets in each dimension of $\mathcal{A}_{L_t}$. The aggregated action $\tilde{a}_{L_t} = [\tilde{O}_m, \tilde{O}_n, \tilde{M}_k]$ is then mapped by a single-layer MLP to the embedding space of $h_O$ and $h_M$, resulting in the historical action embedding $h_a \in \mathbb{R}^q$.

### 4.3.4 Decision Making

Given that machine assignments are determined in FJSP solutions (i.e., for any action $[O_m, O_n, M_k]$, $O_n$ must be processed by $M_k$), the third dimension can be omitted from the action space. We enhance each operation embedding $h_{Oij}$ by concatenating $h_M$ and $h_a$, then feed them into the policy network to obtain intermediate vectors forming a matrix $F_O \in \mathbb{R}^{|\mathcal{O}| \times q}$. We multiply $F_O$ by its transpose to obtain a score matrix $F_{OO}$ of shape $|\mathcal{O}| \times |\mathcal{O}|$, where each element represents the priority score of an action in $\mathcal{A}_t$. Based on the Nopt2 neighborhood, we mask infeasible actions by setting their scores to $-\infty$ and apply softmax normalization to obtain the action probability distribution. By leveraging the FJSP constraints, we reduce the space complexity of $\mathcal{A}_t$ from $O(|\mathcal{O}|^2|\mathcal{M}|)$ to $O(|\mathcal{O}|^2)$, which alleviates the difficulty of learning and accelerates model convergence. A detailed analysis of this space complexity reduction is provided in Appendix E.

### 4.3.5 Training Algorithm

We propose an $n$-step PPO algorithm with a parallel greedy exploration strategy to efficiently explore the large solution space of FJSP. At each step $t$, the policy samples $P$ candidate actions

Table 1 Results on small synthetic instances

| Size | | Sample | | Greedy | | 100 Iterations | | | | 200 Iterations | | | | 400 Iterations | | | | OR-Tools[1] |
|---|---|---|---|---|---|---|---|---|---|---|---|---|---|---|---|---|---|---|
| | | DANIEL | HGNN | DANIEL | HGNN | GD | FI | BI | MIStar | GD | FI | BI | MIStar | GD | FI | BI | MIStar | |
| **SD1** 10×5 | Obj. | 101.67 | 105.59 | 106.76 | 111.67 | **100.00** | 100.91 | 100.55 | 100.02 | 100.00 | 100.91 | 100.54 | **99.88** | 100.00 | 100.91 | 100.53 | **99.69** | |
| | Gap | 5.55% | 9.62% | 10.84% | 15.94% | **3.82%** | 4.77% | 4.39% | 3.84% | 3.82% | 4.77% | 4.38% | **3.70%** | 3.82% | 4.77% | 4.37% | **3.50%** | 96.32(5%) |
| | Time[2] | 0.26s | 0.44s | 0.16s | 0.17s | 8.13s | 12.60s | 12.09s | 5.64s | 16.43s | 27.54s | 26.21s | 11.23s | 32.95s | 58.71s | 54.73s | 23.00s | |
| 20×5 | Obj. | 192.78 | 207.53 | 197.56 | 211.22 | 191.31 | 191.79 | 191.36 | **190.87** | 191.31 | 191.79 | 191.36 | **190.79** | 191.31 | 191.79 | 191.36 | **190.67** | |
| | Gap | 2.46% | 10.30% | 5.00% | 12.26% | 1.68% | 1.93% | 1.71% | **1.45%** | 1.68% | 1.93% | 1.71% | **1.40%** | 1.68% | 1.93% | 1.71% | **1.34%** | 188.15(0%) |
| | Time | 1.14s | 1.05s | 0.33s | 0.34s | 53.86s | 70.73s | 77.32s | 15.72s | 1.80m | 2.46m | 2.70m | 33.38s | 3.60m | 5.20m | 5.56m | 1.16m | |
| 15×10 | Obj. | 153.22 | 160.86 | 161.28 | 166.92 | 151.96 | 152.10 | 151.90 | **150.85** | 151.96 | 152.10 | 151.90 | **150.61** | 151.96 | 152.10 | 151.90 | **150.34** | |
| | Gap | 6.75% | 12.07% | 12.37% | 16.30% | 5.87% | 5.97% | 5.83% | **5.10%** | 5.87% | 5.97% | 5.83% | **4.93%** | 5.87% | 5.97% | 5.83% | **4.74%** | 143.53(7%) |
| | Time | 1.90s | 2.17s | 0.50s | 0.50s | 58.62s | 86.43s | 90.21s | 27.09s | 1.97m | 3.08m | 3.11m | 54.22s | 3.93m | 6.81m | 6.33m | 1.80m | |
| 20×10 | Obj. | 193.91 | 214.81 | 198.50 | 215.78 | 192.92 | 193.26 | 193.03 | **192.71** | 192.92 | 193.26 | 193.03 | **192.70** | 192.92 | 193.26 | 193.03 | **192.68** | |
| | Gap | -1.06% | 9.61% | 1.29% | 10.10% | -1.56% | -1.39% | -1.51% | **-1.67%** | -1.56% | -1.39% | -1.51% | **-1.67%** | -1.56% | -1.39% | -1.51% | **-1.68%** | 195.98(0%) |
| | Time | 2.71s | 2.93s | 0.68s | 0.69s | 2.44m | 2.90m | 3.09m | 32.91s | 4.90m | 6.06m | 6.64m | 1.14m | 9.86m | 12.83m | 13.87m | 2.51m | |
| **SD2** 10×5 | Obj. | 365.26 | 479.37 | 413.73 | 552.80 | 360.17 | 354.22 | 349.66 | **345.05** | 360.17 | 354.22 | 349.66 | **343.43** | 360.17 | 354.22 | 349.66 | **342.50** | |
| | Gap | 11.96% | 46.94% | 26.82% | 69.45% | 10.40% | 8.58% | 7.18% | **5.77%** | 10.40% | 8.58% | 7.18% | **5.27%** | 10.40% | 8.58% | 7.18% | **4.98%** | 326.24(96%) |
| | Time | 0.30s | 0.22s | 0.16s | 0.20s | 6.06s | 10.37s | 11.74s | 5.27s | 12.21s | 22.48s | 25.65s | 10.33s | 24.39s | 48.51s | 54.57s | 21.39s | |
| 20×5 | Obj. | 629.56 | 960.11 | 671.38 | 1047.83 | 626.35 | 622.75 | 619.13 | **615.12** | 626.35 | 622.75 | 619.13 | **613.75** | 626.35 | 622.75 | 619.13 | **612.73** | |
| | Gap | 4.57% | 59.48% | 11.52% | 74.05% | 4.04% | 3.44% | 2.84% | **2.17%** | 4.04% | 3.44% | 2.84% | **1.95%** | 4.04% | 3.44% | 2.84% | **1.78%** | 602.04(0%) |
| | Time | 0.78s | 0.77s | 0.33s | 0.35s | 48.48s | 62.37s | 72.23s | 11.55s | 1.61m | 2.18m | 2.52m | 23.50s | 3.30m | 4.71m | 5.21m | 47.93s | |
| 15×10 | Obj. | 522.51 | 757.18 | 588.72 | 824.62 | 516.79 | 511.00 | 514.73 | **488.22** | 516.79 | 511.00 | 514.73 | **476.93** | 516.79 | 511.00 | 514.73 | **465.71** | |
| | Gap | 38.53% | 100.75% | 56.09% | 118.63% | 37.02% | 35.48% | 36.47% | **29.44%** | 37.02% | 35.48% | 36.47% | **26.45%** | 37.02% | 35.48% | 36.47% | **23.47%** | 377.17(28%) |
| | Time | 1.72s | 1.45s | 0.50s | 0.57s | 60.34s | 89.27s | 78.09s | 18.09s | 2.07m | 3.17m | 2.64m | 37.42s | 4.19m | 6.87m | 5.30m | 1.31m | |
| 20×10 | Obj. | 552.38 | 987.57 | 605.37 | 1036.65 | 549.30 | 542.39 | 535.06 | **530.93** | 549.30 | 542.39 | 535.06 | **528.92** | 549.30 | 542.39 | 535.06 | **525.49** | |
| | Gap | 19.01% | 112.76% | 30.42% | 123.34% | 18.34% | 16.85% | 15.27% | **14.39%** | 18.34% | 16.85% | 15.27% | **13.95%** | 18.34% | 16.85% | 15.27% | **13.21%** | 464.16(1%) |
| | Time | 2.84s | 2.91s | 0.69s | 0.71s | 2.35m | 3.06m | 3.53m | 25.76s | 4.64m | 7.07m | 7.54m | 53.25s | 9.14m | 15.33m | 15.76m | 1.88m | |

[1] For OR-Tools, the solution and the ratio of optimally solved instances are reported.
[2] "s", "m", and "h" denote seconds, minutes, and hours, respectively.

$\{a_t^1, a_t^2, \ldots, a_t^P\}$ from $A_t$, evaluates their improvements in parallel, and selects the action $a_t^i$ yielding the greatest makespan reduction for execution. This approach enables the exploration of $P$ solutions per iteration, producing high-quality solutions in fewer iterations and significantly reducing search time. More detailed analysis and pseudo-code of our training algorithm are presented in Appendix F.

## 5 Experiments

In this section, we present the experimental setup, performance evaluations on synthetic and public datasets as well as the results of ablation studies.

### 5.1 Experimental Settings

**Datasets.** We evaluate our method on both synthetic data and benchmarks. The synthetic datasets SD1 [8] and SD2 [22] cover 6 scales and 100 instances per scale, and differ in distribution. For each $n \times m$ FJSP instance, the number of compatible machines $|\mathcal{M}_{ij}|$ for $O_{ij}$ is sampled from $U(1, m)$. In SD1, the number of operations per job and the average processing time $\bar{p}_{ij}$ of $O_{ij}$ are sampled from $U(0.8 \times m, 1.2 \times m)$ and $U(1, 20)$, respectively. The machine-specific processing time $p_{ij}^k$ is drawn from $U(0.8 \times \bar{p}_{ij}, 1.2 \times \bar{p}_{ij})$. While in SD2, each job has $m$ operations, and $p_{ij}^k$ is sampled from $U(1, 99)$. Benchmarks include Hurink [54] (Edata, Rdata, Vdata; 40 instances per set over 8 scales) and Brandimarte [55] (10 instances across 7 scales). Implementation details are in Appendix G.

**Baselines and Performance Metrics.** We compare our method with two DRL-based construction methods, HGNN [8] and DANIEL [22]. HGNN is a heterogeneous GNN designed to encode features represented by a heterogeneous disjunctive graph, while DANIEL employs a dual-attention network on tight graph representation. Both models are retrained and tested, with results reported for sampling and greedy strategies. We also compare three hand-crafted improvement rules: greedy(GD) [46], best-improvement (BI), and first-improvement (FI) [56]. Specifically, GD selects the solution with the smallest makespan from the neighborhood, while BI and FI choose the best and first improving ones, respectively. A *restart* strategy [57] is applied in BI and FI to escape local optima by continuing the search from a new initial solution. We also compares with (near-)optimal solutions from Google OR-Tools, a powerful constraint programming solver, under a 1800 s time limit. For benchmarks, we report the results of a two-stage genetic algorithm (2SGA) [58] directly from [22].

As such incremental search methods cannot quickly access very different solutions [27, 42], their performance is sensitive to the initial solution quality. We use DANIEL to sample 100 initial solutions per instance, which are equally applied to three rule-based methods for fair comparison. *MIStar* then performs parallel iterative search, retaining the best solution. Performance is measured by the average makespan and the relative gap to the best-known solution, which is obtained via OR-Tools for synthetic datasets [22] and from [59] for public benchmarks.

Table 2 Generalization results on large synthetic instances

| Size | | Sample | | Greedy | | 100 Iterations | | | | 200 Iterations | | | | 400 Iterations | | | | OR-Tools[1] |
|---|---|---|---|---|---|---|---|---|---|---|---|---|---|---|---|---|---|---|
| | | DANIEL | HGNN | DANIEL | HGNN | GD | FI | BI | MIStar | GD | FI | BI | MIStar | GD | FI | BI | MIStar | |
| **SD1** 30×10 | Obj. | 286.77 | 308.55 | 288.61 | 314.71 | 286.04 | 286.19 | 285.95 | **285.79** | 286.04 | 286.19 | 285.95 | **285.75** | 286.04 | 286.19 | 285.95 | **285.68** | |
| | Gap | 4.41% | 12.33% | 5.08% | 14.58% | 4.14% | 4.19% | 4.11% | **4.05%** | 4.14% | 4.19% | 4.11% | **4.03%** | 4.14% | 4.19% | 4.11% | **4.01%** | 274.67(6%) |
| | Time[2] | 5.60s | 6.25s | 0.97s | 0.97s | 7.67m | 10.51m | 10.73m | 57.03s | 15.32m | 22.00m | 21.49m | 1.83m | 30.65m | 47.10m | 43.11m | 3.77m | |
| 40×10 | Obj. | 379.71 | 410.76 | 379.28 | 417.87 | 379.04 | 379.05 | 378.92 | **378.82** | 379.04 | 379.05 | 378.92 | **378.76** | 379.04 | 379.05 | 378.92 | **378.63** | |
| | Gap | 3.76% | 12.24% | 3.64% | 14.18% | 3.57% | 3.58% | 3.54% | **3.51%** | 3.57% | 3.58% | 3.54% | **3.50%** | 3.57% | 3.58% | 3.54% | **3.46%** | 365.96(3%) |
| | Time | 9.94s | 10.45s | 1.30s | 1.34s | 18.20m | 25.13m | 24.04m | 1.11m | 36.48m | 52.01m | 48.46m | 2.55m | 1.22h | 1.82h | 1.63h | 5.94m | |
| **SD2** 30×10 | Obj. | 756.52 | 1453.40 | 800.02 | 1536.74 | 753.66 | 748.41 | 739.47 | **735.14** | 753.66 | 748.41 | 739.47 | **731.84** | 753.66 | 748.41 | 739.47 | **728.76** | |
| | Gap | 9.28% | 109.95% | 15.57% | 121.99% | 8.87% | 8.11% | 6.82% | **6.19%** | 8.87% | 8.11% | 6.82% | **5.72%** | 8.87% | 8.11% | 6.82% | **5.27%** | 692.26(0%) |
| | Time | 5.76s | 5.98s | 0.99s | 1.14s | 7.65m | 12.13m | 12.16m | 49.43s | 15.04m | 24.58m | 25.49m | 1.69m | 29.50m | 51.78m | 52.62m | 3.56m | |
| 40×10 | Obj. | 953.14 | 1937.96 | 984.55 | 2045.78 | 947.64 | 941.60 | 930.44 | 931.04 | 947.64 | 941.60 | 930.44 | **929.13** | 947.64 | 941.60 | 930.44 | **925.93** | |
| | Gap | -4.53% | 94.11% | -1.39% | 104.91% | -5.08% | -5.69% | **-6.81%** | -6.75% | -5.08% | -5.69% | -6.81% | **-6.94%** | -5.08% | -5.69% | -6.81% | **-7.26%** | 998.39(0%) |
| | Time | 10.06s | 11.10s | 1.34s | 1.36s | 18.29m | 25.07m | 21.06m | 73.29s | 35.83m | 54.84m | 42.18m | 2.52m | 1.18h | 1.89h | 1.42h | 5.40m | |

[1] For OR-Tools, the solution and the ratio of optimally solved instances are reported.
[2] "s", "m", and "h" denote seconds, minutes, and hours, respectively.

Table 3 Results on public benchmarks

| Method | mk | | | la(rdata) | | | la(edata) | | | la(vdata) | | |
|---|---|---|---|---|---|---|---|---|---|---|---|---|
| | Obj. | Gap | Time(s) | Obj. | Gap | Time(s) | Obj. | Gap | Time(s) | Obj. | Gap | Time(s) |
| OR-Tools | 174.20 | 0.99% | 1447.08 | 935.80 | 0.16% | 1397.43 | 1028.93 | 0.01% | 899.60 | 919.60 | -0.01% | 639.17 |
| 2SGA | 175.20 | 1.57% | 57.60 | - | | | - | | | 812.20[1] | 0.39% | 51.43 |
| DANIEL(S)[2] | 180.80 | 4.81% | 1.71 | 984.63 | 5.39% | 1.99 | 1120.28 | 8.88% | 2.00 | 934.75 | 1.64% | 2.02 |
| HGNN(S) | 192.20 | 11.42% | 1.72 | 1004.60 | 7.53% | 1.84 | 1123.10 | 9.16% | 2.44 | 934.23 | 1.58% | 2.12 |
| DANIEL(G) | 184.30 | 6.84% | 0.45 | 1044.85 | 11.84% | 0.48 | 1176.40 | 14.34% | 0.48 | 965.38 | 4.97% | 0.48 |
| HGNN(G) | 200.00 | 15.94% | 0.46 | 1035.75 | 10.86% | 0.52 | 1189.33 | 15.59% | 0.48 | 962.48 | 4.66% | 0.47 |
| GD-100[3] | 178.92 | 3.72% | 22.86 | 963.03 | 3.08% | 26.09 | 1100.46 | 6.96% | 4.14 | 924.53 | 0.53% | 89.94 |
| FI-100 | 178.85 | 3.68% | 34.91 | 968.60 | 3.67% | 36.16 | 1101.43 | 7.05% | 6.77 | 927.33 | 0.83% | 131.90 |
| BI-100 | 178.43 | 3.44% | 37.98 | 964.80 | 3.27% | 41.25 | 1100.63 | 6.97% | 5.91 | 927.55 | 0.86% | 122.02 |
| MIStar-100 | **177.80** (±0.74)[4] | **3.07%** | 17.50 | **958.90** (±4.55) | **2.64%** | 19.86 | **1099.38** (±4.17) | **6.85%** | 19.60 | **923.55** (±1.81) | **0.42%** | 20.36 |
| GD-200 | 178.92 | 3.72% | 45.73 | 963.03 | 3.08% | 51.94 | 1100.46 | 6.96% | 8.27 | 924.53 | 0.53% | 179.99 |
| FI-200 | 178.85 | 3.68% | 74.64 | 968.60 | 3.67% | 76.17 | 1101.43 | 7.05% | 14.45 | 927.33 | 0.83% | 272.34 |
| BI-200 | 178.43 | 3.44% | 81.03 | 964.65 | 3.25% | 86.20 | 1100.60 | 6.97% | 12.70 | 926.55 | 0.75% | 251.11 |
| MIStar-200 | **177.70** (±0.75) | **3.01%** | 36.45 | **957.23** (±4.72) | **2.46%** | 40.66 | **1099.30** (±4.13) | **6.84%** | 40.05 | **922.70** (±1.71) | **0.33%** | 41.83 |
| GD-400 | 178.92 | 3.72% | 91.79 | 963.03 | 3.08% | 103.81 | 1100.46 | 6.96% | 16.51 | 924.53 | 0.53% | 360.28 |
| FI-400 | 178.85 | 3.68% | 162.16 | 968.60 | 3.67% | 158.27 | 1101.35 | 7.04% | 31.56 | 927.33 | 0.83% | 586.32 |
| BI-400 | 178.43 | 3.44% | 169.70 | 964.50 | 3.24% | 180.45 | 1100.33 | 6.94% | 27.24 | 925.55 | 0.64% | 521.21 |
| MIStar-400 | **177.60** (±0.63) | **2.96%** | 76.35 | **956.38** (±4.04) | **2.37%** | 85.45 | **1099.18** (±4.25) | **6.83%** | 83.77 | **921.85** (±1.77) | **0.24%** | 87.23 |

[1] The makespan and gap of 2SGA on la(vdata) benchmark are computed on la1-30 instances, as reported in [58].
[2] "(S)" denotes the sampling strategy, and "(G)" denotes the greedy strategy.
[3] "-100", "-200", and "-400" denote different iteration budgets.
[4] Values are reported as the average (± standard deviation) over 10 independent runs.

## 5.2 Performance on Synthetic Instances

We train our model with 100 iterations on four small sizes ($10 \times 5$, $20 \times 5$, $15 \times 10$, and $20 \times 10$) from two synthetic datasets and evaluate on corresponding test instances. Table 1 reports the performance and average runtime per instance across all methods. The generalization capability of the model, evaluated on large-sized instances ($30 \times 10$ and $40 \times 10$ in Table 2) and with extended iterations up to 400, is also reported. The results show that *MIStar* consistently outperforms two construction methods across various instances, with larger gains on SD2. This is because initial solutions on SD1 are near-optimal, leaving limited room for improvement, as verified by experiments on varying-quality initial solutions (see Appendix I). Compared with rule-based improvement heuristics, *MIStar* achieves better solution quality with shorter runtime and maintains consistent improvement over longer iterations, verifying the effectiveness and efficiency of its policy. See detailed analysis in Appendix H.

## 5.3 Performance on Public Benchmarks

We evaluate generalization on Hurink and Brandimarte benchmarks using the model trained on the $10 \times 5$ instances from SD2. While OR-Tools and 2SGA achieve near-optimal results, their computational cost limits practicality. *MIStar* maintains stable performance over both DRL-based construction and rule-base method across all benchmarks, demonstrating its effective generalization through the learned general policy. Regarding solving time, rule-based methods exhibit notable variation across Hurink dataset, with runtimes increasing from `edata` to `rdata` and `vdata` due to the growing set of assignable machines that enlarge the neighborhood. In contrast, *MIStar* maintains stable runtime across all distributions.

Table 4 Generalization Performance on Larger-Scale Instances

| Instance Size | Method | Objective | Time (min) | Optimality Gap |
|---|---|---|---|---|
| 50×15 | OR-Tools | 881.45 | 60.0 | 39.05% |
| (LB:537.2) | MIStar-200 | 969.7 | 10.6 | 44.60% |
| 60×15 | OR-Tools | 1075.1 | 60.0 | 46.94% |
| (LB:570.2) | MIStar-200 | 1109.8 | 14.2 | 48.62% |
| 100×10 | OR-Tools | 1922.65 | 60.0 | 62.22% |
| (LB:726.5) | MIStar-200 | 2138.4 | 15.9 | 66.03% |
| 50×30 | OR-Tools | Infeasible | 60.0 | - |
| (LB:N/A) | MIStar-200 | 1040.1 | 60.0 | - |

## 5.4 Results on Larger Instances

To further evaluate the generalization and scalability of our framework, we conducted experiments on even larger instances (up to 1,500 operations). For each problem size, 10 instances were randomly generated, and both *MIStar* and OR-Tools were given a one-hour time limit per instance. We report their average optimality gap across different scales, whcih is calculated based on the Lower Bound (LB) as $\left(1 - \frac{\text{LB}}{\text{Objective}}\right) \times 100\%$. For *MIStar*, the results were measured at 200 iterations using the model trained on instances of size $20 \times 15$.

The results in Table 4 highlight two critical advantages of our approach. First and most strikingly, on the highly complex $50 \times 30$ instances, OR-Tools failed to produce any feasible solution within the time limit, whereas *MIStar* consistently found high-quality solutions. This demonstrates *MIStar*'s superior robustness and scalability on truly challenging problems. Second, for the other large instances where OR-Tools did find a solution, *MIStar* achieves comparable solution quality in a small fraction of the time. For example, on the $100 \times 10$ instances, *MIStar* reaches a $66.03\%$ optimality gap in just under 16 minutes, while OR-Tools requires a full hour to achieve a $62.22\%$ gap. This massive speed-up, combined with the strong generalization from a model trained only on smaller instances, confirms that *MIStar* provides an effective and highly scalable solution for large-scale FJSP.

## 5.5 Ablation Studies

We conducted ablation experiments on the SD2 dataset to assess the memory module and parallel greedy search strategy. As evidenced in Table 7 (Appendix J), their combination improves performance: the parallel greedy search strategy significantly enhances search efficiency and mitigates the risk of early convergence to local optima. Meanwhile, the memory module optimizes policy decisions, leading to higher solution quality. We further investigated the impact of the parallel scale $P$ on our strategy. Figure 8 (Appendix J) demonstrates that increasing $P$ generally improves performance at the cost of longer runtimes, with the optimal value depending on the instance characteristics. Therefore, $P$ should be properly adjusted to achieve a trade-off between runtime and solution quality.

# 6 Conclusion

This study proposes *MIStar*, the first DRL-based improvement heuristic framework for solving the FJSP. By formulating the iterative refinement as an MDP, we introduce a heterogeneous disjunctive graph representation tailored for FJSP solutions and employ a memory-enhanced heterogeneous graph neural network for feature extraction and thorough exploration of the solution space. Furthermore, we propose a parallel greedy search strategy that significantly reduces the number of iterations while achieving high-quality solutions. Extensive experiments on synthetic data and public benchmarks demonstrate the superior performance of *MIStar* over both traditional handcrafted improvement heuristics and DRL-based construction methods. Future work may focus on enabling the network to adaptively adjust the parallel scale $P$, and on extending our methodology to disruptive local moves in larger neighborhoods, such as the reconstruction of solution segments, to further enhance exploration and reduce sensitivity to initial solutions.

## Acknowledgements

This work is supported by the Jilin Provincial Department of Science and Technology Project under Grant Grant20240101369JC.

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

# A  Comparative Analysis of Neighborhood Structures

The critical path, whose length represents the makespan $C_{\max}$, is essential in constructing feasible solutions [55]. Operations on this path are critical operations, and critical blocks are maximal sequences of adjacent critical operations processed on the same machine [45]. In local search, the neighborhood comprises feasible solutions generated by small perturbations to the current one [55]. For scheduling, these perturbations typically involve reordering operations on the critical path. The neighborhood structure directly affects the efficiency of the search algorithm, making it important to eliminate unnecessary or infeasible moves [45]. Among various neighborhood designs, the N5 neighborhood is significantly smaller than others [45]. It identifies a single critical path and defines a move by reversing either the first two or the last two operations within a critical block.

While effective for JSP [21], the N5 neighborhood cannot be applied to FJSP. Its limitation lies in only permitting reordering within one critical block, where operations are processed on the same machine. This perturbation preserves the original machine assignments, severely constraining the exploration of the solution space. More critically, unlike JSP where operations of the same job are processed on different machines, FJSP allows multiple operations of a job to be processed on the same one. Consequently, swapping operations within critical blocks in FJSP may violate job precedence constraints, resulting in infeasible solutions (Figure 5).

In contrast, we employ the Nopt2 neighborhood [51] that simultaneously modifies operation sequences and machine assignments while maintaining feasibility through time-constrained reinsertion (see paragraph 3). This structure enables broader solution space exploration and improves efficiency by using optimal insertion intervals to reduce the neighborhood size.

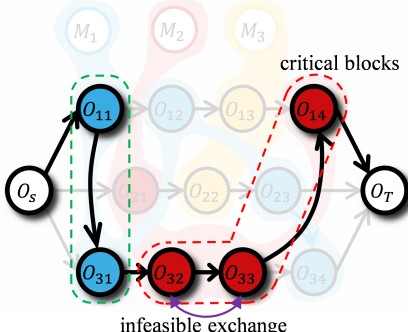

Figure 5 **Infeasible exchange caused by the N5 neighborhood in FJSP.** Dotted lines of different colors are critical blocks.

# B  Definition of Raw Feature Vectors

The raw features of operations and machines at each state $s_t$ (omitting step $t$) are defined as follows. *Features of Operations*: Each operation node $O_{ij} \in \mathcal{O} \setminus \{O_S, O_T\}$ is represented by a 9-dimensional feature vector $\mathbf{x}_{O_{ij}}$:

1) **Minimum Processing Time**: shortest processing time among compatible machines.
2) **Average Processing Time**: mean processing time among compatible machines.
3) **Range of Processing Time**: difference between the maximum and minimum processing times among compatible machines.
4) **Machine Availability Ratio**: ratio of the number of compatible machines to total machines.
5) **Processing Time**: processing time on the assigned machine.
6) **Earliest Start Time**: earliest possible start time of the operation.
7) **Latest Start Time**: latest start time without schedule delay.
8) **Scheduling Order on Machine**: execution order on machine $M_k$ (starting from 1).
9) **Processing Time of Job**: total processing time of parent job.

*Features of Machines*: Each machine node $M_k \in \mathcal{M}$ owns a 4-dimensional feature vector $\mathbf{y}_{M_k}$:

1) **Utilization Rate**: ratio of active processing time to makespan.

2) **Criticality Degree**: proportion of operations on critical path among processable operations.
3) **Load Ratio**: percentage of assigned operations versus machine capacity.
4) **Processing Efficiency**: mean processing time of assigned operations.

## C   Advantages of the Directed Heterogeneous Graph

In the FJSP, the global scheduling state comprises both operations and machines. However, the absence of machine nodes in traditional disjunctive graphs results in an indirect representation of machine information through numerical values [21], which limits the policy from assigning operations to available machines effectively. The heterogeneous disjunctive graph $\mathcal{H}$ [8] addresses this by introducing machine nodes (Fig. 6), where each operation connects to compatible machines via undirected edges, such that solving the FJSP is equivalent to selecting one O-M arc per operation while removing others. This reduces the graph density from $\sum_{k=1}^{|\mathcal{M}|} \binom{n_k}{2}$ to $\sum_{k=1}^{|\mathcal{M}|} n_k$ with $n_k$ being the number of operations assignable to machine $M_k$ and enables better injection of machine status [8].

While beneficial for construction methods, this structure provides limited advantages for improvement approaches where complete solutions contain fixed machine assignments. This leads to a total of $|\mathcal{O}|$ disjunctive arcs—more than the $|\mathcal{O}| - |\mathcal{M}|$ arcs in the conventional graph, negating the benefit of reduced graph density. Most importantly, undirected edges hinder the clear encoding of machine processing sequences, which is crucial for distinguishing between different scheduling solutions and identifying the critical path when constructing the action space [21, 51]. Instead, our heterogeneous graph $\overrightarrow{\mathcal{H}}$(Figure 1c), tailored for complete schedules, overcomes this limitation by encoding machine processing sequences through directed hyper-edges, which effectively represent the ordering of operations across different solutions without significantly increasing graph density. It adopts a unified structure with fully directed arrows, avoiding the complexity of mixed edge types and better aligning with current learning-to-search graph-based frameworks [21].

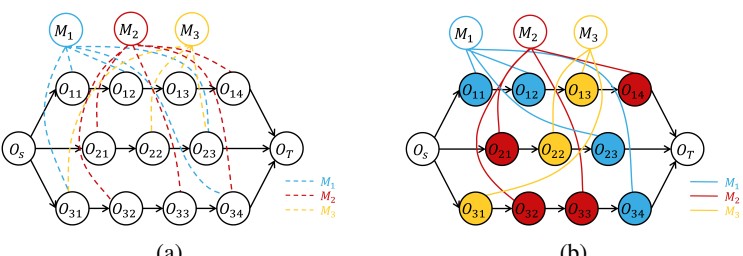

(a)                                                    (b)

Figure 6 **Graph representations in $\mathcal{H}$ [8].** (a) FJSP instance; (b) feasible solution.

## D   Compact State Representation in Memory

### D.1   Motivation and Compact Formulation

During the improvement, we store the state-action pair $(s_t, a_t)$ at each step in a memory module. However, the state is represented as a complex heterogeneous graph with rich node attributes and adjacency relations, leading to excessive memory consumption if fully stored. Furthermore, evaluating similarity between such graphs typically requires GNNs to extract embeddings, introducing additional network components and computational cost.

To address this, we simplify the graph representation to better support our memory mechanism. First, since our current goal is to distinguish different scheduling solutions—rather than using node features as heuristic guidance—we omit the node attributes and retain only the graph topology. Second, for a given problem instance, solutions differ only in operation sequences on machines (i.e., hyper-arcs), while sharing the same job precedence constraints (i.e., conjunctive arcs). Based on this insight, we isolate the hyper-arcs from $\overrightarrow{\mathcal{H}}_t$ (see Fig. 7) and use them as the key feature to characterize each solution. Specifically, we use the operation-machine incidence matrix $\boldsymbol{L}_t$ as a compact representation of $s_t$ and measure similarity via the Frobenius inner product.

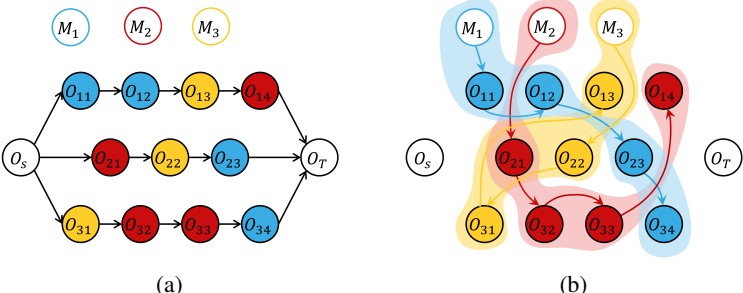

Figure 7 **Decomposition of FJSP solution structure.** (a) shared job precedence constraints; (b) distinct machine processing sequences.

Instead of serving as merely binary indicators of machine assignment, the non-zero element $(\boldsymbol{L}_t)_{i,j}$ represents the processing order index of operation $i$ on its assigned machine $j$. This crucial design ensures that two schedules with identical machine assignments but different sequences (e.g., a good sequence vs. a poor one) are correctly distinguished. For instance, a good sequence of operations corresponding to [*1, 2, 3, 4, 5*] and a poor, reversed sequence [*5, 4, 3, 2, 1*] on the same machine will be recognized as maximally dissimilar by the Frobenius inner product. The mathematical foundation for this lies in the **Rearrangement Inequality**, which guarantees that the inner product is maximized for similarly ordered sequences and minimized for oppositely ordered ones. This allows our similarity metric to be both computationally efficient and highly sensitive to the quality of the operation sequence.

**Theorem 1.** *The Rearrangement Inequality states that, for two sequences* $a_1 \le a_2 \le \cdots \le a_n$ *and* $b_1 \le b_2 \le \cdots \le b_n$, *the inequalities*

$$a_1 b_n + a_2 b_{n-1} + \cdots + a_n b_1 \le a_1 b_{\pi(1)} + a_2 b_{\pi(2)} + \cdots + a_n b_{\pi(n)} \le a_1 b_1 + a_2 b_2 + \cdots + a_n b_n$$

*hold, where* $\pi(1), \pi(2), \ldots, \pi(n)$ *is any permutation of* $1, 2, \ldots, n$.

### D.2 Alternative Designs

We have considered and tested other expressive representations for the historical states that explicitly encode sequence information.

**Multi matrices:** The adjacency matrix $A_t^M$ of operations on machines encodes the processing order between operations, while the binary operation-machine incidence matrix $L_t^B$ indicates machine assignment. The state similarity is then calculated by computing the inner products of both matrices separately and summing the results, expressed as:

$$\omega_{t,t'} = \langle L_t^B, L_{t'}^B \rangle_F + \langle A_t^M, A_{t'}^M \rangle_F \tag{8}$$

**Difference matrix:** For the operation-machine incidence matrices that store processing order indices, we compute the element-wise difference between two schedules, $L_t$ and $L_{t'}$, to obtain a difference matrix, and then use the reciprocal of its Frobenius norm to measure similarity, formulated as:

$$\omega_{t,t'} = \frac{1}{|L_t - L_{t'}|_F + \epsilon} = \frac{1}{\sqrt{\sum_{i=1}^{|\mathcal{O}|} \sum_{j=1}^{|\mathcal{M}|} |(L_t)_{i,j} - (L_{t'})_{i,j}|^2} + \epsilon} \tag{9}$$

However, these alternatives introduced higher storage or computational overhead without yielding significant performance gains. Our chosen representation thus strikes a deliberate and effective balance between representational power and efficiency.

## E    Complexity Analysis of Action Space

Let $F_O \in \mathbb{R}^{|\mathcal{O}| \times q}$ and $F_M \in \mathbb{R}^{|\mathcal{M}| \times q}$ denote the embedding matrices of operation and machine nodes, respectively. To assign a priority score to each possible triple-action $[O_m, O_n, M_k]$, we compute the priority matrix $F_{OOM} \in \mathbb{R}^{|\mathcal{O}| \times |\mathcal{O}| \times |\mathcal{M}|}$ through a two-step tensor operations involving $F_O$ and $F_M$. As a result, the space complexity of the action space $\mathcal{A}_{original}$ is $O(|\mathcal{O}|^2 |\mathcal{M}|)$. Prior

work [60] has shown that a large number of possible actions is known to lead to over-optimistic estimates of future rewards, degrading agent performance and learning efficiency. To address this, we reshape the action space by exploiting the constraint of FJSP: each operation can only be assigned to a unique machine. Given a solution of FJSP, there exists a mapping $f_M : \mathcal{O} \to \mathcal{M}$, which allows us to omit the third dimension and represent actions as pairs $[O_m, O_n]$. The space complexity of $\mathcal{A}_{optimized}$ is reduced to $O(|\mathcal{O}|^2)$ and the optimized priority matrix is calculated as follows:

$$F_{OO} = F_O F_O^\top \in \mathbb{R}^{|\mathcal{O}| \times |\mathcal{O}|} \tag{10}$$

We can formalize the relationship between the original and optimized action spaces as follows:

**Theorem 2.** *The optimized action space $\mathcal{A}_{optimized}$ is functionally equivalent to $\mathcal{A}_{original}$ in terms of admissible actions under the FJSP constraints.*

*Proof.* For any feasible action $a = [O_m, O_n, M_k] \in \mathcal{A}_{original}$, there exists a unique $a' = [O_m, O_n] \in \mathcal{A}_{optimized}$ with $M_k = f_M(O_n)$. Conversely, any $a' \in \mathcal{A}_{optimized}$ can be mapped back to $a \in \mathcal{A}_{original}$ via $f_M$. Thus, this compaction preserves all feasible actions while eliminating invalid ones (i.e., actions where the operation and machine do not match). This shaping of the action space reduces learning difficulty and accelerates model convergence without compromising the agent's performance. □

---

**Algorithm 1:** Training procedure with $n$-step PPO using parallel search.

---

**Input:** *MIStar* with trainable parameters $\Theta = \{\delta, \theta, \varphi\}$, parallel scale $P$, iteration limit $T$, update size $n$, validate size $v$, new data generation step $d$, batch size $B$, total number of training epochs $I$;

**Output:** Trained *MIStar* with parameters $\Theta^* = \{\delta^*, \theta^*, \varphi^*\}$;

1   **for** $i = 0$ **to** $i < I$ **do**
2     **if** $i \bmod d = 0$ **then**
3       Randomly generate $B$ instances;
4     **end**
5     Compute initial solutions $\{s_0^1, \ldots, s_0^B\}$ using DANIEL;
6     **for** $t = 0$ **to** $T$ **do**
7       **for** $s_t^b \in \{s_t^1, \ldots, s_t^B\}$ **do**
8         Initialize a training data buffer $D^b$ and memory buffer $H^b$ with size 0;
9         Extract embeddings using MHGNN;
10        **for** $p = 1$ **to** $P$ **do**
11          Sample a local move $a_t^{bp} \sim \pi_\delta(\cdot | s_t^b)$;
12          Compute $C_{\max}(s_{t+1}^{bp})$ w.r.t $a_t^{bp}$;
13        **end**
14        Identify the best action $a_t^{b*}$ with greatest improvement ;
15        Update $s_t^b$ w.r.t $a_t^{b*}$ and compute similarity score $\Omega$ from $H^b$;
16        Compute reward $r_t(s_t^b, a_t^{b*}) = r_{gain} - r_{penalty}$;
17        Store the data $(s_t^b, a_t^{b*}, r_t)$ into $D^b$ and store $(s_t^b, a_t^{b*})$ into $H^b$;
18        **if** $t \bmod n = 0$ **then**
19          Compute the generalised advantage estimates $\hat{A}_t$;
20          Compute the PPO loss $L$, and optimize the parameters $\Theta$ for $R$ epochs;
21          Update network parameters;
22        **end**
23        Clear buffers $D^b$ and $H^b$;
24       **end**
25     **end**
26     **if** $i \bmod v = 0$ **then**
27       Validate the policy;
28     **end**
29     $i = i + B$;
30 **end**

---

# F  The $n$-step PPO Algorithm with Parallel Greedy Exploration Strategy

We adopt the PPO algorithm [61] based on the actor-critic architecture as our training strategy. The actor is the policy network and the critic provides state value estimation. The network parameters are updated every $n$ steps, which effectively addresses sparse rewards and out-of-memory issues [21]. The flexibility of FJSP leads to a large solution space, requiring numerous iterations to converge to high-quality solutions [30]. To address this challenge, we introduce a parallel greedy exploration strategy, which evaluates multiple candidate actions in parallel and greedily selects the one yielding the greatest improvement. This approach enables the exploration of multiple solutions per iteration, yielding high-quality results with fewer iterations. Moreover, by prioritizing solutions with the greatest improvement, the model is guided toward more promising paths, minimizing the interference from inefficient paths during gradient updates and accelerating convergence.

The pseudo-code of the $n$-step PPO algorithm is presented in Algorithm 1.

# G  Training Configurations

All hyperparameters were fine-tuned on the smallest-scale instances from SD2 and kept the same for all other models. Experiments were conducted on a platform equipped with an Intel i9-12900K processor with 24 cores, 64 GB memory, and an NVIDIA RTX 4090 GPU with 24 GB VRAM. Detailed specifications are provided in Table 5. Upon acceptance of this manuscript for publication, the full code will be made available on GitHub.



Table 5 Training configuration parameters

| Parameter | Value |
|---|---|
| Number of MHGNN layers ($L$) | 4 |
| Number of Actor network layers | 4 |
| Number of Critic network layers | 3 |
| Number of attention heads in GAT ($n_H$) | 1 |
| Dimension of hidden layers ($q$) | 64 |
| GIN learnable parameter ($\epsilon$) | 0 |
| Discount factor ($\gamma$) | 1 |
| Clipping ratio | 0.2 |
| Policy function coefficient | 1 |
| Value function coefficient | 0.5 |
| Entropy coefficient | 0.01 |
| Learning rate | $5 \times 10^{-4}$ |
| Optimizer | Adam |
| Memory buffer size | 600 |
| KNN retrieval count ($K$) | 15 |
| Similarity penalty coefficient ($\lambda$) | 10 |
| PPO update interval ($n$) | 10 |
| Rounds per PPO update ($R$) | 3 |
| Total training epochs ($I$) | 20,000 |
| Validation interval ($v$) | 5 |
| Instance generation interval ($d$) | 20 |
| Iterations per instance ($T$) | 100 |
| Batch size ($B$) | 20 |
| Parallel scale ($P$) | 50 |



# H  Detailed Analysis on Synthetic Instances

Our experiments on four small-sized instances from SD2 reveal that *MIStar* generates an action space consisting of dozens to hundreds of possible moves per iteration using the Nopt2 neighborhood—up to 100 times larger than [21] for solving JSP. While this expanded action space provides greater

Table 6 Results of different initialization methods

| Method | Iter. | 10×5 | | 20×5 | | 15×10 | | 20×10 | |
|---|---|---|---|---|---|---|---|---|---|
| | | Obj. | Gap | Obj. | Gap | Obj. | Gap | Obj. | Gap |
| DANIEL(S) | 0 | 365.26 | 0.00% | 629.56 | 0.00% | 522.51 | 0.00% | 552.38 | 0.00% |
| | 100 | 345.05 | 5.53% | 615.12 | 2.29% | 488.22 | 6.56% | 530.93 | 3.88% |
| | 200 | 343.43 | 5.98% | 613.75 | 2.51% | 476.93 | 8.72% | 528.92 | 4.25% |
| | 400 | **342.50** | 6.23% | **612.73** | 2.67% | **465.71** | 10.87% | **525.49** | 4.87% |
| HGNN(S) | 0 | 479.35 | 0.00% | 963.03 | 0.00% | 759.54 | 0.00% | 984.64 | 0.00% |
| | 100 | 375.20 | 21.73% | 745.18 | 22.62% | 677.96 | 10.74% | 879.48 | 10.68% |
| | 200 | 366.35 | 23.57% | 702.90 | 27.01% | 626.90 | 17.46% | 812.53 | 17.48% |
| | 400 | 359.10 | 25.09% | 674.93 | 29.92% | 569.81 | 24.98% | 737.42 | 25.11% |
| SPT | 0 | 503.86 | 0.00% | 799.56 | 0.00% | 670.56 | 0.00% | 775.03 | 0.00% |
| | 100 | 399.25 | 20.76% | 692.43 | 13.40% | 578.48 | 13.73% | 695.34 | 10.28% |
| | 200 | 393.74 | 21.86% | 680.62 | 14.88% | 550.94 | 17.84% | 673.44 | 13.11% |
| | 400 | 389.34 | 22.73% | 671.05 | 16.07% | 521.96 | 22.16% | 648.48 | 16.33% |
| MWKR | 0 | 493.36 | 0.00% | 949.85 | 0.00% | 737.44 | 0.00% | 953.11 | 0.00% |
| | 100 | 382.09 | 22.55% | 747.12 | 21.34% | 664.69 | 9.87% | 858.45 | 9.93% |
| | 200 | 372.62 | 24.47% | 705.15 | 25.76% | 618.86 | 16.08% | 799.61 | 16.11% |
| | 400 | 364.99 | 26.02% | 676.88 | 28.74% | 563.56 | 23.58% | 729.91 | 23.42% |
| FIFO | 0 | 476.96 | 0.00% | 931.50 | 0.00% | 763.12 | 0.00% | 978.03 | 0.00% |
| | 100 | 376.29 | 21.11% | 729.85 | 21.65% | 669.51 | 12.27% | 861.06 | 11.96% |
| | 200 | 367.70 | 22.91% | 693.35 | 25.57% | 618.74 | 18.92% | 799.15 | 18.29% |
| | 400 | 360.68 | 24.38% | 668.26 | 28.26% | 562.59 | 26.28% | 728.91 | 25.47% |
| RANDOM | 0 | 585.08 | 0.00% | 1067.38 | 0.00% | 1062.39 | 0.00% | 1304.70 | 0.00% |
| | 100 | 382.24 | 34.67% | 750.69 | 29.67% | 733.17 | 30.99% | 960.68 | 26.37% |
| | 200 | 371.38 | 36.52% | 707.33 | 33.73% | 661.03 | 37.78% | 865.85 | 33.64% |
| | 400 | 365.33 | **37.56%** | 678.00 | **36.48%** | 589.53 | **44.51%** | 772.15 | **40.82%** |

optimization potential, it also introduces more suboptimal solutions, increasing the risk of local optima entrapment. Moreover, even for small-sized problems, OR-Tools could only optimally solve a small portion of instances within the time limit [22]. These observations underscore the complexity of the FJSP. Compared with rule-based improvement heuristics, *MIStar* shows superior performance under the same iteration budget, with only a marginal shortfall (0.06%) against the BI rule on the SD2 $40 \times 10$ instance at 100 iterations. Notably, even equipped with *restart*, rule-based methods still tend to stagnate as iterations increase—likely due to the large and complex solution space—whereas *MIStar* continues to improve, demonstrating the effectiveness of the learned policy. Additionally, *MIStar* reduces runtime by directly outputting local moves, avoiding exhaustive evaluations across the entire neighborhood.

# I    Improvement over Varied Initial Solution Generation Methods

We systematically evaluate the optimization capabilities of *MIStar* using six initialization strategies: two DRL-based construction methods with sampling (DANIEL(S) and HGNN(S)), random initialization, and three classical priority dispatching rules—shortest processing time (SPT), most work remaining (MWKR), and first-in-first-out (FIFO) [55]. Experiments are conducted on the SD2 dataset with models trained on instances of the same size, to assess the performance across different initialization methods under the fixed iteration budget. Improvement is measured by the makespan gap, calculated as $\left(1 - \frac{C_{\text{best}}}{C_0}\right) \times 100\%$, where $C_0$ is the makespan of the initial solution and $C_{\text{best}}$ is the incumbent optimal value.

The results are detailed in Table 6. It can be observed that the quality of initial solutions impacts the search efficiency of *MIStar*. With the same number of iterations, better initial solutions enable *MIStar* to generate higher-quality schedules, while poorer ones typically allow for larger relative improvements. Notably, after 400 iterations, solutions initialized by HGNN(S), MWKR, and FIFO outperform those generated by DANIEL(S) with 0 iteration on the $10 \times 5$ instances. This demonstrates that, with a sufficient iteration budget, *MIStar* has the potential to elevate weaker initial solutions to a performance level comparable to that of stronger ones. Moreover, even when starting from (near)-optimal solutions initialized by DANIEL(S), *MIStar* still achieves notable improvements, further validating the effectiveness of our framework.

Table 7 Performance of ablated models

| Model | Iter. | 10×5 | | 20×5 | | 15×10 | | 20×10 | |
|---|---|---|---|---|---|---|---|---|---|
| | | Obj. | Time[1] | Obj. | Time | Obj. | Time | Obj. | Time |
| Baseline | 100 | 364.27 | 0.76s | 629.09 | 1.16s | 519.34 | 1.82s | 552.46 | 2.37s |
| | 200 | 364.27 | 1.48s | 629.09 | 2.44s | 519.34 | 3.69s | 552.46 | 4.78s |
| | 400 | 364.27 | 2.89s | 629.09 | 4.85s | 519.34 | 7.25s | 552.46 | 9.48s |
| w/o GIN | 100 | 348.31 | 4.97s | 616.71 | 10.72s | 491.78 | 18.12s | 548.61 | 24.80s |
| | 200 | 347.06 | 10.78s | 614.98 | 22.88s | 484.36 | 37.58s | 545.97 | 53.68s |
| | 400 | 345.73 | 20.56s | 613.98 | 45.90s | 477.80 | 1.45m | 544.31 | 1.91m |
| w/o GAT | 100 | 350.86 | 5.82s | 618.67 | 11.34s | 492.76 | 17.85s | 548.0 | 22.63s |
| | 200 | 349.89 | 9.46s | 617.33 | 22.91s | 486.74 | 36.97s | 547.97 | 51.02s |
| | 400 | 348.33 | 21.41s | 615.99 | 46.79s | 480.66 | 1.29m | 547.87 | 1.78m |
| w/o HGAT | 100 | 350.49 | 5.44s | 618.15 | 11.39s | 491.97 | 17.95s | 535.78 | 23.89s |
| | 200 | 349.08 | 9.74s | 616.56 | 23.01s | 483.46 | 37.01s | 532.83 | 52.47s |
| | 400 | 347.17 | 20.10s | 614.97 | 46.38s | 473.83 | 1.29m | 529.92 | 1.67m |
| w/o Par. | 100 | 363.02 | 0.59s | 627.20 | 0.98s | 518.28 | 1.68s | 551.39 | 2.41s |
| | 200 | 363.02 | 1.22s | 627.20 | 2.13s | 518.28 | 4.02s | 551.39 | 6.18s |
| | 400 | 363.02 | 2.59s | 627.20 | 5.05s | 518.28 | 11.72s | 551.39 | 17.57s |
| w/o Mem. | 100 | 346.11 | 5.47s | 615.99 | 13.98s | 490.63 | 17.25s | 534.31 | 25.47s |
| | 200 | 344.76 | 11.04s | 615.32 | 25.33s | 486.16 | 35.85s | 531.71 | 51.08s |
| | 400 | 343.53 | 20.24s | 614.80 | 48.91s | 482.16 | 73.14s | 528.44 | 1.69m |
| Ours | 100 | 345.05 | 5.27s | 615.12 | 11.55s | 488.22 | 18.09s | 530.93 | 25.76s |
| | 200 | 343.43 | 10.33s | 613.75 | 23.50s | 476.93 | 37.42s | 528.92 | 53.25s |
| | 400 | **342.50** | 21.39s | **612.73** | 47.93s | **465.71** | 1.31m | **525.49** | 1.88m |

[1] "s" and "m" denote seconds and minutes, respectively.

## J    Results on Ablation Studies

### J.1    Analysis on Components and Strategy

We conducted comprehensive ablation studies to evaluate the contributions of the key components of our framework, including both the model architecture and the search strategy. The experiments were performed on four small-size instances from the SD2 dataset.

We first assessed the roles of the memory module and the parallel greedy search strategy. Three ablated variants are compared: the *Baseline* model without any enhancements, a version without the parallel greedy search strategy (*w/o Par.*), and another without the memory module (*w/o Mem.*). Results in Table 7 demonstrate the critical roles of both components. Although our parallel approach takes longer per iteration, it achieves superior solutions in fewer steps, improving search efficiency. In contrast, the model without parallelism runs faster but consistently converges to inferior local optima. Furthermore, while the memory module helps enhance the decision-making of the policy, it alone (i.e., *w/o Par.*) proves insufficient to escape local optima. This limitation may stem from the fact that the FJSP solution space is extremely large and complex, making it challenging for the policy network to achieve effective global exploration through sparse, single-step sampling. For future work, a critical direction involves dynamically constraining the search space or optimizing the memory mechanism to selectively extract salient historical features.

Our tailored representation module, which is composed of multiple specialized graph encoders, is not only theoretically motivated by the structural characteristics of FJSP graphs but is also aligned with mainstream graph learning practices [8, 21, 24, 50]. The ablation results by removing each component (i.e., *w/o GAT*, *w/o GIN* and *w/o HGAT*) confirm that each of them is essential for learning informative and discriminative state representations that support effective policy learning.

### J.2    Evaluation of Search Scale

Using models trained with $P = 50$, we systematically evaluated the search strategy across different parallel scales $P \in \{10, 20, 30, 40, 50, 60, 70\}$ on SD2 instances of two sizes. The result curves are illustrated in Figure 8. It is evident that increasing $P$ leads to longer runtimes under a fixed number of iterations. Moreover, the runtime scales linearly with the iteration count across different $P$ values. Regarding solution quality, the 10×5 instance has a relatively small neighborhood (approximately 80 solutions), making $P = 20$ sufficient to cover most promising moves. Consequently, further increasing $P$ does not yield noticeable improvement. On the other hand, the 15×10 instance has

around 240 neighbors, so increasing $P$ from 10 to 60 continuously improves performance. However, the difference between $P = 60$ and $P = 70$ becomes negligible, indicating a saturation point beyond which additional evaluations offer little benefit. These observations highlight the importance of selecting an appropriate $P$ based on the problem size and the flexibility of alternative machines to balance performance gains and computational cost. In future work, we plan to enable the network to dynamically adapt $P$ according to instance characteristics, reducing manual tuning efforts.

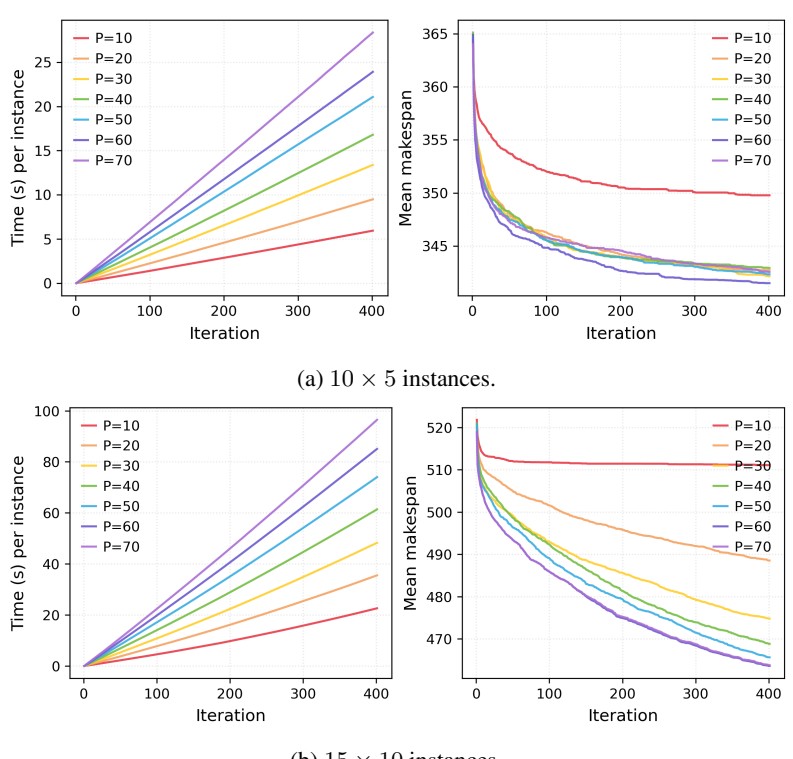

(a) $10 \times 5$ instances.

(b) $15 \times 10$ instances.

Figure 8 **Runtime (left) and performance (right) curves across different parallel scale.**

