# OpenReview forum: "Learning Memory-Enhanced Improvement Heuristics for Flexible Job Shop Scheduling"
_NeurIPS.cc/2025/Conference — NeurIPS 2025 poster_

### Official Review · Reviewer_W83L · 2025-07-01

**Clarity:** 3
**Significance:** 2
**Originality:** 2
**Rating:** 4
**Confidence:** 3

**Summary:**

This paper proposes a Deep Reinforcement Learning (DRL)-based improvement heuristic framework named MIStar to solve FJSP problems. The model is presented to address (1) complex state representation by a novel heterogeneous disjunctive graph, (2) effective policy learning by the proposed heterogeneous graph representation learning, and (3) efficient search strategies by a parallel greedy search strategy.

**Questions:**

Please refer to the above weaknesses.

**Ethical Concerns:**

["NO or VERY MINOR ethics concerns only"]

**Final Justification:**

The author’s motivation seems more like improving the original technique in order to refresh the baseline metrics, but ignoring the needs in practical applications.

The author showed the model’s robustness through comprehensive evaluation metrics, but did not initially consider the “From and Back” problem. FJSP is a task driven by industrial needs, yet I have doubts about whether this work can, in turn, be applied back to industrial applications. Accordingly, I provided my concerns and suggestions towards this.
I appreciate the author’s descriptive answers to several of my questions, but based on my practical experience, I still keep my concerns on the feasibility of the algorithm in industrial applications, particularly regarding whether the model can correctly understand the attributes such as machine functions, changes in quantity, and disable period. This should be proven by more experiments, which do not have any large dataset support to date. Therefore, based on the author’s experimental results and analysis, I have retained my Rating of 4.

**Limitations:**

Please refer to the above weaknesses.

**Paper Formatting Concerns:**

No major formatting concerns.

**Quality:**

2

**Strengths And Weaknesses:**

Strengths:
1. The integration of memory mechanisms with heterogeneous graph neural networks is innovative and well-suited for FJSP.
2. The proposed method outperforms the traditional heuristics and DRL-based constructive methods across both synthetic and real-world benchmarks.

Weakness:
1. The analysis of embedding is not presented, even the authors claim that the embeddings are significant for memorizing historical data and extracting state features.
2. It is unclear whether the GIN and GAT are de facto significant and effective for topological information encoding and semantic information enhancing respectively, referring to many advanced GNN models. More analysis related to the module selection in operation node embedding should be provided.
3. It is very common to witness changes in the functionality of devices (some devices are capable of doing two kinds of actions), alterations in the quantity of equipment, and sudden shifts in the priority of the job-shops in real smart factories. Although this is a technical work for solving FJSP only, I would cordially encourage the authors to discuss if MIStar can address those challenges or if the authors will consider these in the future works.
4. Although the answer to open access was No, I would encourage the authors to open source the code or provide detailed instructions for dataset generation and modeling.

---

> ### Author Rebuttal · Authors · 2025-07-31
>
> We appreciate the reviewer for their valuable feedback and address the main concerns below:
> ## Analysis of embedding
> In our framework, the policy network makes decisions based on both the current scheduling state (operation and machine embeddings) and historical information (historical action embedding). Specifically，
>
> - **Operation embeddings** capture the current solution state. Since different scheduling solutions correspond to different graph structures[1]—i.e., different adjacency relations among operation nodes—we extract the topological structure of the graph and the contextual dependencies[1] among operations as the operation embeddings.
> - **Machine embeddings** reflect the machine load by aggregating the features of neighboring operation nodes, and serve as critical heuristics for the policy to reassign operations from overloaded machines to less loaded ones[2], thereby reducing the makespan.
> - **Historical action embeddings** represent actions taken under similar historical states. They serve as a reference to encourage diverse action selection and enhance exploration[3], when combined with the similarity penalty in the reward function.
>
> ## Module selection in operation
> Topological information and the semantic information of constraints are key to distinguishing different operation embeddings[1].
> - **For topological information**, which distinguishes different schedule structures, we chose **GIN**[4]. Its proven strength, equivalent to the Weisfeiler-Lehman test, is essential for capturing the subtle but critical structural differences between operation nodes in varying solutions[1].
> - **For semantic information**, we recognized that an operation's neighbors—job predecessors and machine predecessors—carry different semantic meanings[1, 2]. To handle this, we employed the **GAT**[6]. Its attention mechanism is perfectly suited to learn the varying semantic roles and importance of these different neighbor types, assigning weights accordingly and thus enriching the node representations.
>
> Our module selection is not only theoretically motivated by the structural characteristics of FJSP graphs, but also aligned with mainstream practices[1, 2, 5, 7]. We conducted ablation studies by removing each component. Results below confirm that each of them is essential for learning informative and discriminative state representations that support effective policy learning.
> |||10×5||20×5||15×10||20×10||
> |-|-|-|-|-|-|-|-|-|-|
> |**Model**|**Iter.**|**Obj.**|**Time**|**Obj.**|**Time**|**Obj.**  |**Time**|**Obj.**|**Time**|
> |**w/o GIN**|100|348.31|4.97s|616.71|10.72s|491.78|18.12s|548.61|24.80s|
> ||200|347.06|10.78s|614.98|22.88s|484.36|37.58s|545.97|53.68s|
> ||400|345.73|20.56s|613.98|45.9s|477.8|1.45m|544.31|1.91m|
> |**w/o GAT**|100|350.86|5.82s|618.67|11.34s|492.76|17.85s|548.0|22.63s|
> ||200|349.89|9.46s|617.33|22.91s|486.74|36.97s|547.97|51.02s|
> ||400|348.33|21.41s|615.99|46.79s|480.66|1.29m|547.87|1.78m|
> |**w/o HGAT**|100|350.49|5.44s|618.15|11.39s|491.97|17.95s|535.78|23.89s|
> ||200|349.08|9.74s|616.56|23.01s|483.46|37.01s|532.83|52.47s|
> ||400|347.17|20.1s|614.97|46.38s|473.83|1.29m|529.92|1.67m|
> |**MIStar**|100|345.05|5.27s|615.12|11.55s|488.22|18.09s|530.93|25.76s|
> ||200|343.43|10.33s|613.75|23.50s|476.93|37.42s|528.92|53.25s|
> ||400|342.50|21.39s|612.73|47.93s|465.71|1.31m|525.49|1.88m|
>
> While more advanced GNNs exist, our aim was to establish a strong, well-justified, and interpretable baseline. We believe this study provides a clear and robust validation of our architectural choices.
> ## Dynamic change challenges
>
> We are grateful for the reviewer's encouragement to discuss MIStar's broader applicability. Although our current work focuses on the standard FJSP, we designed MIStar with future dynamic adaptations in mind. We believe its core architectural strengths—a flexible graph representation, a size-agnostic GNN policy, and an adaptable reward function—make it well-suited to address the challenges mentioned.
> ### 1. Multi-capability Devices
> The scenario where machines are capable of performing multiple types of actions naturally aligns with the modeling paradigm of FJSP. The various operations within a job typically correspond to different stages of processing, each representing a specific type of task[10]. In contrast to JSP, where each operation must be processed by a fixed machine[9](i.e., each machine performs only one type of action), FJSP allows each operation to be processed by multiple candidate machines[10]. This implies that machines can handle multiple processing stages within a job, thereby exhibiting multi-processing capabilities. Therefore, MIStar, as a method for solving FJSP, inherently supports the modeling and optimization of device functionality variability.
> ### 2. Machine quantity changes
> Although changes in machine quantity may alter the graph structure, the size-agnostic nature of our GNN-based policy network enables MIStar to dynamically re-optimize[12] from any valid existing solution, without restarting the scheduling process from scratch. Furthermore, our memory mechanism remains robust, as the incidence matrices from schedules of varying sizes can be aligned via simple padding or cropping before similarity comparison. In addition, the reward function should not only focus on minimizing makespan but also incorporate schedule stability in re-scheduling scenarios. To this end, we can introduce a stability term[11] into the reward to minimize the differences in operation start times between the original and revised schedules $\sum_{j=1}^{|J|} \sum_{i=1}^{n_j} \left| st_t - st_{t-1} \right|$, thus encouraging minimal disruption to the former schedule[11,13].
>
> To further handle different types of equipment changes, we introduce more specific strategies:
>
> **Machine Addition:**
> - New jobs can be integrated by expanding the state matrix and enriching feature vectors with arrival times. Their feature vectors should also be enriched with the arrival time.
> - To avoid interference from irrelevant information, operations completed before the arrival of the new machine will not be recorded in the state. This helps reduce noise and ensures a compact state representation during re-scheduling with a large number of added machines.
>
> **Machine Reduction:**
> - Unavailable machines can be removed by pruning their corresponding entries in the state matrices. Alternatively, the original graph structure can be retained by setting the processing time on failed machines to infinity. In this case, the agent, which aims to minimize the makespan, will naturally avoid assigning operations to a machine marked as unavailable or having an infinite processing time.
> - All operations currently assigned to failed machine must be re-assigned. Broken machines are masked out when generating candidate actions based on the Nopt2 neighborhood structure. As a result, invalid actions $[O_m, O_n, M_k]$ involving broken machines in the third dimension are effectively filtered out.
> ### 3. Shifts in priority
>
> Job priorities may shift due to changes in customer demand, resulting in urgent jobs requiring higher priority[8]. This can be elegantly addressed through the **reward function**. Jobs can be grouped by urgency of customer orders—i.e., normal jobs and urgent jobs. The objective function for this scheduling problem can be augmented as the weighted sum of the makespan for normal jobs and the total tardiness for urgent jobs[8], which is to be minimized. Moreover, it is possible to divide jobs into more than two groups based on finer-grained urgency levels. The weights assigned to each group can be dynamically adjusted according to their priority. To adapt to real-time shifts in job priorities, the grouping and weighting can be periodically updated at fixed iteration intervals during the scheduling process.
>
> ## On Open-Sourcing the code
> We are fully committed to making our code and models publicly available upon acceptance of the paper to facilitate reproducibility and future research in the community.
> ##
> We extend our heartfelt thanks to the reviewer for their detailed and thoughtful observations during the rebuttal process. Integrating these insights with the new experimental data and our discussions, we will enhance the final version of the paper to improve its clarity and presentation.
>
> [1]Deep reinforcement learning guided improvement heuristic for job shop scheduling, 2022.
> [2]Flexible job-shop scheduling via graph neural network and deep reinforcement learning, 2022.
> [3]Marco: A memory-augmented reinforcement framework for combinatorial optimization, 2024.
> [4]How powerful are graph neural networks?, 2018.
> [5]A multi-action deep reinforcement learning framework for flexible Job-shop scheduling problem, 2022.
> [6]Graph attention networks, 2017.
> [7]Graph neural networks for job shop scheduling problems: A survey, 2025.
> [8]Metaheuristics for a flow shop scheduling problem with urgent jobs and limited waiting times, 2021.
> [9]Deterministic job-shop scheduling: Past, present and future, 1999.
> [10]Routing and scheduling in a flexible job shop by tabu search, 1993.
> [11]Dynamic scheduling in flexible job shop systems by considering simultaneously efficiency and stability, 2010.
> [12]A review of dynamic scheduling: context, techniques and prospects, 2021.
> [13]Multiobjective flexible job-shop rescheduling with new job insertion and machine preventive maintenance, 2022.

---

> > ### Comment · Reviewer_W83L · 2025-08-06
> > **Official Comments**
> >
> > Thank you for the rebuttal. I have also carefully reviewed the comments provided by the other reviewers and the authors' responses.
> >
> > The proposed model indeed shows promise in terms of feasibility and robustness, but I still have some concerns regarding its practical applicability in more complex, real-world scenarios, particularly regarding the machine addition and sudden device functionality alteration. Furthermore, while the work is robust, it does not introduce significant theoretical advancements.
> >
> > Therefore, I still maintain my rating of Borderline accept (4).

---

> > > ### Author Response · Authors · 2025-08-06
> > >
> > > Thank you for your thoughtful follow-up comment. We are sincerely grateful for your positive acknowledgement that our **work is robust**—for a method designed to tackle complex scheduling problems, we consider this to be very high praise.
> > >
> > > You have highlighted a key aspect of our work’s identity. We would characterize our contribution as sitting at the crucial **intersection of methodology, performance, and theory.** Our framework is the result of a principled design process where key components are motivated by deep theoretical insights into the problem's structure. For instance, our memory metric is formally justified by the Rearrangement Inequality, and our GNN architecture is a deliberate synthesis of GIN and GAT, chosen for their proven theoretical capabilities. This approach reflects a significant and impactful trend within the broader "Learning to Optimize" community. For a wide array of classic NP-hard problems—from routing challenges like TSP and VRP, to resource allocation tasks like JSP and Bin Packing. By demonstrating *how* to construct such a theoretically-grounded yet high-performance framework, we believe our work provides a solid foundation for future explorations in this area. We agree that research focusing on more foundational theory is also critical, and your feedback has been valuable in shaping our perspective on future work.
> > >
> > > We also wanted to share some deeper reflections on the crucial point you raised: **practical applicability in real-world scenarios.**
> > >
> > > We fully agree that bridging the gap to real industrial needs is the ultimate goal. While the scarcity of public industrial data is a challenge, we designed our framework with the core principles of real-world production in mind.
> > >
> > > A guiding principle for us is that **in a real factory, minimizing disruption is often as important as optimizing metrics** [1]. A full, system-wide reschedule, even if optimal on paper, can be prohibitively expensive due to downtime and adjustment costs [1]. This is where MIStar's architectural strengths become critical, particularly in the scenarios you mentioned:
> > >
> > > Scenario 1: Machine Addition in Practice
> > >
> > > Imagine a scenario where, due to a surge in urgent orders, a new CNC machine is rushed onto the factory floor mid-shift to alleviate a critical bottleneck [2]. The naive approach of a full reschedule is often a non-starter; it would create chaos, forcing adjustments across dozens of unrelated workstations.
> > >
> > > This is where MIStar’s capacity for **surgical, localized re-optimization** provides immense value. Its task is not to solve the whole puzzle again, but to answer a much more practical question: "Given this new resource, how can we best re-route *only the jobs affected by the bottleneck* to it, while leaving the rest of the stable production plan untouched?" Because our GNN-based policy is size-agnostic, it can incorporate the new machine without retraining. More importantly, our improvement-based framework allows it to intelligently explore local moves that leverage this new machine, preserving the overall schedule's stability and integrity.
> > >
> > > Scenario 2: Sudden Device Functionality Alteration
> > >
> > > Consider an advanced manufacturing cell where a reconfigurable robotic arm must switch its end-effector from a welding tool to a gripper. This change is not instantaneous; it involves setup time, calibration, and safety checks [3, 4]. A scheduling system that ignores these real-world overheads will produce infeasible and dangerous plans.
> > >
> > > MIStar can handle this with practical realism. The functionality change is modeled as an update to the machine's capabilities in our graph, and the associated setup time is integrated as a delay. Crucially, MIStar's improvement-based nature allows it to **preserve the global schedule** and focus only on re-sequencing the operations immediately dependent on this single robotic arm. It can intelligently decide whether it's better to wait for the changeover or to re-route a critical operation to another, less-optimal machine to maintain flow. This agility to make localized, cost-aware decisions is what maintains production continuity.
> > >
> > > In essence, the "robustness" you observed is a direct outcome of this design philosophy. MIStar is architected not just to find good solutions, but to adapt to change in a way that is efficient, intelligent, and minimally disruptive—qualities that are paramount for any system intended for real-world deployment.
> > >
> > > Thank you once again for your thorough and highly constructive engagement. It has been invaluable in helping us sharpen both our paper and our future research trajectory.
> > >
> > > [1]Dynamic scheduling in flexible job shop systems by considering simultaneously efficiency and stability, 2010.
> > > [2]Real time production improvement through bottleneck control, 2009.
> > > [3]A review of dynamic scheduling: context, techniques and prospects, 2021.
> > > [4]On-line machine scheduling, 1997.

---

### Official Review · Reviewer_wFrm · 2025-07-01

**Clarity:** 3
**Significance:** 2
**Originality:** 1
**Rating:** 4
**Confidence:** 4

**Summary:**

This paper proposes MIStar, a Deep Reinforcement Learning (DRL) framework for the Flexible Job Shop Scheduling Problem (FJSP). The key idea is to move away from constructive DRL methods, which build a schedule from scratch, and instead focus on improvement-based methods that iteratively refine a complete solution. To this end, the authors formulate the problem as a Markov Decision Process (MDP) where the state is a complete schedule and actions correspond to local moves in the Nopt2 neighborhood.
The main contributions of the paper are:
1. A directed heterogeneous disjunctive graph representation for FJSP solutions, which explicitly includes machine nodes and their processing sequences using hyper-arcs.
2. A Memory-enhanced Heterogeneous Graph Neural Network (MHGNN) that encodes the graph state and leverages a memory of previously visited solutions to enhance decision-making and avoid local optima.
3. A parallel greedy search strategy that evaluates multiple candidate actions at each step to improve search efficiency.

**Questions:**

Questions for the Authors:

The state similarity for the memory module is based on the operation-machine incidence matrix, which ignores the processing order on each machine. Could you please justify this design choice? Have you considered or experimented with more expressive representations for historical states that also encode sequence information, and if so, how did they perform?

Regarding the experimental runtimes reported in Table 1 and Table 2: Are the reported times for the baseline methods (GD, FI, BI) the aggregate time for running searches from 100 different initial solutions? Is the reported time for MIStar for a single search trajectory, or is it also an aggregate? Please clarify how the "search effort" is normalized between MIStar and the rule-based baselines to ensure a fair comparison of both solution quality and runtime.

**Ethical Concerns:**

["NO or VERY MINOR ethics concerns only"]

**Final Justification:**

The rebuttal answers the weaknesses I pointed in my initial review.

**Limitations:**

yes

**Quality:**

2

**Strengths And Weaknesses:**

Strengths

The paper tackles the FJSP, a well-known NP-hard problem with significant industrial relevance. The framing of the problem as an improvement task learned via DRL is a promising direction. The authors correctly identify that most recent DRL work has focused on constructive methods, which often struggle to find high-quality solutions. Shifting the focus to learning improvement heuristics is a valuable contribution to the field of neural combinatorial optimization.
- Well-Motivated Methodological Components: Each component of the proposed MIStar framework is well-motivated by a specific challenge in applying improvement methods to FJSP.

Weaknesses

- Ablation Study Missing: There is no evidence or ablation study provided to isolate the benefit of the memory-enhanced heterogeneous graph neural network (MHGNN) compared to standard GNNs without memory. It is unclear how much the memory module contributes to overall performance. The memory module is a core contribution, but its design raises questions. The state similarity is calculated using the Frobenius inner product of the operation-machine incidence matrix (Lt​). This matrix only captures which machine an operation is assigned to, but not the processing sequence on that machine. It is therefore possible for two schedules with very different makespans (e.g., a good sequence vs. a poor sequence on the same machine) to be considered identical or highly similar by this metric. This seems like a significant oversimplification that could lead to misleading historical information. The paper needs to provide a much stronger justification for why this simplified representation is sufficient.
- The paper employs parallel greedy exploration runs, but it is not clear whether the computational budget is held constant across all compared models. If MIStar is allowed to explore more solutions in parallel, it may have an unfair advantage in solution quality.
- The term "agent" (e.g., "agent selects an action...") is introduced without prior definition, which may reduce clarity for readers unfamiliar with the terminology.
-The descriptions accompanying Table 2 are brief, and the tables themselves are dense and potentially overwhelming. The paper would benefit from additional visualizations to better communicate model comparisons and results.
- The paper does not provide code for reproducing the results, limiting the ability of the community to validate and build upon this work.
Limited Choice of "Hand-crafted" Baselines: While the inclusion of GD, BI, and FI is a good start, they are relatively basic local search algorithms. The field of metaheuristics for FJSP is mature, with highly effective algorithms like Tabu Search (e.g., the original Brandimarte paper ) and Genetic Algorithms.
Incremental Novelty of Graph Representation: The paper introduces a "novel directed heterogeneous disjunctive graph". However, it is explicitly "inspired by the heterogeneous disjunctive graph H [8]", which is the graph representation used by the HGNN baseline.

---

> ### Author Rebuttal · Authors · 2025-07-31
>
> We thank the reviewer for their valuable suggestions and comments and address the main concerns below:
> ## On the Memory Mechanism: Ablation Study and Design Justification
> ### 1. ablation study
>
> We would like to clarify that an ablation study on the memory module was indeed conducted and is presented in Appendix I (Table 5). To directly answer the reviewer's concern, **this study demonstrates that MIStar consistently outperforms its variant without the memory module, with performance gains ranging from 0.41% to 4.36% across different problem scales.** This confirms the significant and positive contribution of our memory-enhanced design.
>
> ### 2. design of the state similarity metric
> We appreciate the reviewer's sharp observation, which gives us an opportunity to clarify a subtle but crucial detail of our design. We commit to adding more details of $L_t$ to the revised version. The reviewer correctly notes that ignoring sequence information would be a major flaw. However, our operation-machine incidence matrix, $L_t$, **does encode sequence information**. Its elements are not merely binary indicators (0/1) of machine assignment, but rather represent the *processing order index* of each operation on its assigned machine.
>
> This design ensures that two schedules with identical machine assignments but different sequences (e.g., a good sequence vs. a poor one) are correctly distinguished. For instance, a good sequence `[1, 2, 3, 4, 5]` and a poor, reversed sequence `[5, 4, 3, 2, 1]` on the same machine will be recognized as maximally dissimilar by the Frobenius inner product. The mathematical foundation for this lies in the **Rearrangement Inequality**, which guarantees that the inner product is maximized for similarly ordered sequences and minimized for oppositely ordered ones. This allows our similarity metric to be both computationally efficient and highly sensitive to the quality of the operation sequence.
>
> **Theorem**: *The Rearrangement Inequality states that, for two sequences*
> $a_1 \le a_2 \le \dots \le a_n$ *and* $b_1 \le b_2 \le \dots \le b_n$, *the inequalities*
>
> $$
> a_1 b_1 + a_2 b_2 + \dots + a_n b_n \ge a_1 b_{\pi(1)} + a_2 b_{\pi(2)} + \dots + a_n b_{\pi(n)} \ge a_n b_1 + a_{n-1} b_2 + \dots + a_1 b_n
> $$
>
> *hold, where* $\pi(1), \pi(2), \dots, \pi(n)$ *is any permutation of* $1, 2, \dots, n$.
>
> ### 3. alternative designs
> We have considered and tested other expressive representations for the historical states that explicitly encode sequence information. For example:
> 1. The adjacency matrix $A_t^{M}$ of operations on machines clearly encodes the processing order between operations, while the binary operation-machine incidence matrix $L_t^{B}$ indicates machine assignment. The state similarity is then calculated by computing the inner products of both matrices separately and summing the results, expressed as:
>
> $$
> \begin{align*}
> \omega_{t,t'} = \langle L_t^{B}, L_{t'}^{B} \rangle_F + \langle A_t^{M}, A_{t'}^{M} \rangle_F
> \end{align*}
> $$
>
> 2. For the operation-machine incidence matrices that store processing order indices, we compute the element-wise difference between two schedules, $L_t$ and $L_{t'}$, to obtain a difference matrix, and then use the reciprocal of its Frobenius norm to measure similarity, formulated as:
>
> $$
> \omega\_{t,t'} = \frac{1}{\| L\_t - L\_{t'} \|\_F + \epsilon} = \frac{1}{\sqrt{\sum\_{i=1}^{\mathcal{O}} \sum\_{j=1}^{\mathcal{M}} |(L\_{t})\_{ij} - (L\_{t'})\_{ij})|^2} + \epsilon} $$
>
> However, these alternatives introduced higher storage and computational overhead without yielding significant performance gains. Our chosen representation thus strikes a deliberate and effective balance between representational power and efficiency.
>
> ## Fairness of Parallel Exploration
>
> MIStar circumvents exhaustive neighborhood evaluation and its overall computational budget is notably smaller than that of the baselines.
>
> Typically, the policy network samples an action from the neighborhood to generate a new schedule. The *parallel greedy exploration* refers to the process where, at each step, the policy network samples P(e.g., 50) actions and evaluates the resulting 50 candidate solutions in parallel, selecting the one with the best improvement for further refinement. However, the baseline methods (GD, FI, BI) evaluate the entire neighborhood at each step and select the “optimal” (e.g., FI selects the first improved solution while BI selects the best one) for the next iteration[1]. In this sense, these baselines can also be considered as performing "parallel" evaluations on a larger scale, since P in their case equals the full neighborhood size.
>
> For example, in a 15×10 SD2 instance with an Nopt2 neighborhood size of 240, MIStar evaluates only 50 candidates per step, whereas the baselines evaluate all 240. Thus, our overall computational cost is significantly lower.
>
> ## Runtime Clarity
> For each instance, we generate 100 initial solutions using a sampling strategy, which together form a batch for parallel improvement. The final result for the instance is selected as the best solution within this batch. The reported runtime is the average time per instance after the iterative process finishes, obtained by dividing the total (aggregate) time by the batch size. The runtimes reported in Table 1 and Table 2 for both MIStar and the rule-based baselines (GD, FI, BI) follow this same setting to ensure consistency.
>
> To ensure a fair comparison, we define the “search effort” as the number of iterations. That is, at a fixed number of iteration steps (e.g., at step 100), we record the corresponding solution quality and runtime for both MIStar and the baselines.
>
> ## Improving Readability
> Thanks for your careful proofreading. We will gladly revise the manuscript to include a definition of the "agent," add clarifying annotations and visualizations for our tables.
>
> * To improve clarity, we will add a brief definition of *“agent”* in Section 4.1, such as: *“agent, the decision-maker based on a given policy, selects an action…”*.
>
> * To address the concern regarding the brief descriptions of Table 2, we will add clarifying footnotes to explain key notations, as follows:
>   1. “-100”, “-200”, and “-400” denote different iteration budgets;
>   2. “Gap” refers to the average gap to the best-known solutions;
>   3. “Time(s)” denotes the average time for solving a single instance in seconds.
> * We will revise the table layout in the final version and include box plots to better visualize model comparisons and results.
>
> ## On the Choice of Baselines
>
> We thank the reviewer for this suggestion. We agree that classic metaheuristics like Tabu Search and Genetic Algorithms are powerful and highly effective solvers for the FJSP.
>
> Our work, however, is situated within the specific and growing research paradigm of **learning-based improvement heuristics**. The primary goal of this paradigm is not to outperform a highly-tuned, problem-specific metaheuristic, but rather to **learn a generalizable policy** that can discover effective improvement strategies automatically and adapt to different problem instances without manual redesign.
>
> Therefore, we chose to focus our main comparison on two classes of baselines: (1) other state-of-the-art **learning-based methods**[1,2,4], which represent the most direct competitors within our research area, and (2) fundamental local search heuristics (GD, FI, BI), which serve to clearly isolate and demonstrate the value added by our learned policy. We believe this comparison provides the clearest picture of our contribution to the field of *neural combinatorial optimization*. We consider a deep comparison with highly-tuned traditional metaheuristics an exciting avenue for future work.
> ## Code
> We are fully committed to releasing our code upon the acceptance of the paper to support reproducibility and facilitate future research.
>
> ## Novelty of graph representation
> We acknowledge the inspiration from HGNN[2]. However, HGNN is designed for *construction* methods, and its structure provides limited advantages (Appendix B) for *improvement* approaches where complete solutions contain fixed machine assignments[1]. Most importantly, the undirected edges in HGNN hinder the clear encoding of machine processing sequences, which is crucial for distinguishing between different schedules and identifying the critical path when constructing the action space[3].
>
> Instead, our directed heterogeneous graph, tailored for complete schedules, overcomes this limitation by encoding machine processing sequences through directed hyper-edges. These effectively represent the ordering of operations across different solutions without significantly increasing graph density. It adopts a unified structure with fully directed arrows, avoiding the complexity of mixed edge types.
>
> ##
> We deeply thank reviewer’s constructive feedback on strengthening our theoretical foundation. Leveraging these inputs, as well as the additional discussions and analyses, we will refine the final version to improve clarity and ensure a more robust presentation.
>
> [1]Zhang C, Cao Z, Song W, et al. Deep reinforcement learning guided improvement heuristic for job shop scheduling[J]. arXiv preprint arXiv:2211.10936, 2022.
> [2]Song W, Chen X, Li Q, et al. Flexible job-shop scheduling via graph neural network and deep reinforcement learning[J]. IEEE Transactions on Industrial Informatics, 2022, 19(2): 1600-1610.
> [3]Mastrolilli M, Gambardella L M. Effective neighbourhood functions for the flexible job shop problem[J]. Journal of scheduling, 2000, 3(1): 3-20.
> [4]Zhang X, Zhu G Y. A literature review of reinforcement learning methods applied to job-shop scheduling problems[J]. Computers & Operations Research, 2025, 175: 106929.

---

> ### Author Response · Authors · 2025-08-07
>
> Dear reviewer,
>
> As the discussion period is nearing its end, with **fewer than three days left**, I want to ensure we have addressed all your concerns to your satisfaction. If there are any additional points or feedback you'd like us to consider, please let us know at your earliest convenience. Your insights are invaluable to us, and we're eager to address any remaining issues promptly to further improve our work.
>
> Thank you for your time and effort in reviewing our paper.

---

> > ### Comment · Reviewer_wFrm · 2025-08-08
> >
> > Thank you for your rebuttal. I will update my score.

---

> > > ### Author Response · Authors · 2025-08-09
> > >
> > > Thank you for your time and for your positive reconsidering. We sincerely appreciate your constructive feedback and will be sure to incorporate the points from our discussion into the final version.

---

### Official Review · Reviewer_YPJJ · 2025-07-02

**Clarity:** 3
**Significance:** 3
**Originality:** 3
**Rating:** 5
**Confidence:** 4

**Summary:**

This paper presents MIStar, a novel DRL-based improvement heuristic framework for solving the Flexible Job Shop Scheduling Problem (FJSP). Unlike prior approaches that primarily rely on constructive methods, MIStar starts from a complete schedule and iteratively refines it using learned improvement heuristics. Key innovations include: 1) A heterogeneous disjunctive graph representation that models operation sequences and machine assignments explicitly. 2) A memory-enhanced heterogeneous graph neural network (MHGNN) that utilizes past scheduling experiences to improve decision making and reduce local optima traps. 3) A parallel greedy search strategy for efficient exploration of the solution space. Experiments on synthetic and public benchmark datasets demonstrate that MIStar outperforms state-of-the-art DRL-based constructive methods and handcrafted improvement heuristics in both quality and efficiency.

**Questions:**

How would MIStar adapt to real-time job arrivals or unexpected machine unavailability? The current formulation assumes a fixed problem instance. Please clarify whether the memory mechanism and graph representation can support online updates or dynamic re-optimization.

Can the authors provide a reward curve during training? Understanding how quickly the policy converges and how often it gets stuck in local optima would strengthen the interpretation of results and help in reproducing or fine-tuning the model.

Why was no ablation on embedding architecture included? Multiple GNN modules are used, including GIN and GAT. How crucial are each of these components? A component-wise breakdown would clarify their individual contributions.

**Ethical Concerns:**

["NO or VERY MINOR ethics concerns only"]

**Final Justification:**

Thank you for the updates and rebuttals. Since the quality has improved significantly, I change my score.

**Limitations:**

No support for unexpected events: The current model assumes a static schedule and does not consider dynamic changes such as new job insertions or maintenance events. This may limit deployment in real-world systems where dynamic events are common.

No error bars in reported metrics: Performance results lack standard deviation or confidence intervals, making robustness unclear.

**Paper Formatting Concerns:**

No.

**Quality:**

3

**Strengths And Weaknesses:**

The proposed method is well-motivated and carefully implemented. The combination of memory-augmented GNNs and improvement heuristics is novel and technically sound.

The methodology is presented in a well-structured manner, with comprehensive figures, examples, and ablation studies to support understanding.

FJSP is a critical real-world problem, and improving its solution quality has significant industrial implications. This work contributes a viable learning-based improvement framework with promising generalization capabilities.

To the best of my knowledge, this is the first work that brings memory-enhanced GNNs into DRL-based improvement heuristics for FJSP, addressing challenges that existing methods often overlook.

While strong comparisons are made against DRL-based constructive and rule-based improvement methods, the absence of comparisons with classic or recent dynamic scheduling algorithms limits the broader applicability assessment of MIStar.

The framework assumes static environments. In realistic smart manufacturing, events like new job insertions or machine breakdowns are common but not considered or simulated in experiments.

Embedding choices: The paper uses GIN and GAT for operation nodes and a hetero-GAT for machine embeddings, but ablation studies on alternative embedding strategies or modules (e.g., Transformer variants) are missing.

The reward function is based on makespan improvement and redundancy penalties, but no detailed reward curve or sensitivity analysis is presented, making it difficult to assess stability.

While performance is reported in tables, error bars or statistical confidence measures are absent, making reproducibility and robustness harder to judge.

---

> ### Author Rebuttal · Authors · 2025-07-31
>
> We appreciate the reviewers' feedback and address each comment below:
> ## Dynamic events
> We thank the reviewer for this important point. We would like to clarify that our focusing on static environments is not an oversight of dynamic problems, but aligns with the mainstream research paradigm in learning-based scheduling[1,2,3]. Even the static, deterministic FJSP remains a formidable challenge, and developing effective heuristics for it is an active and critical area of research[4].
>
> That being said, we fully agree that adaptability to realistic dynamics is crucial. A key strength of MIStar is its modular architecture, designed for future extensibility and minimal modification to handle dynamic environments. Below, we detail how the graph representation, memory mechanism, and action space can be adapted for two representative examples.
>
> **Handling real-time job arrivals**:
>
> MIStar is exceptionally well-suited for dynamic re-optimization, a key advantage of our improvement-based approach over constructive methods.
> 1. **Graph representation**: The flexibility of our graph structure allows new jobs to be seamlessly integrated by expanding the state matrices to include their operations and associated constraints. To maintain focus and efficiency, completed operations can be pruned from the state, ensuring the agent always reasons over the most relevant part of the problem[3].
> 2. **Memory mechanism**: 0ur memory mechanism remains robust, as the incidence matrices from schedules of different sizes can be aligned via simple padding or cropping before similarity comparison.
> 3. **Action space**: Time-based constraints are easily handled. To respect arrival times, we simply filter out any action that would schedule a job's first operation before it is ready[5]. The problem's inherent precedence constraints, already encoded in our graph, automatically handle the sequencing for all subsequent operations.
>
> **Handling unexpected machine breakdowns**:
> 1. **Graph representation**: Machine status is a feature in our graph representation, allowing the breakdown to be modeled by updating the node. The agent, trained to minimize makespan, will naturally avoid assigning operations to unavailable machines or those with infinite processing time. Alternatively, Unavailable machines can be directly removed from the graph by pruning their corresponding entries in the state matrices.
> 4. **Memory mechanism**: Machine breakdowns can be reflected by assigning a special value (e.g., –1) to unavailable columns in incidence matrices. Alternatively, memory entries can include labels (e.g., breakdown presence or broken machine count) for coarse-grained filtering, followed by fine-grained similarity matching.
> 5. **Action space**: Operations on failed machines must be reassigned. Unavailable machines are masked out in the Nopt2 neighborhood to filter invalid actions $[O_m, O_n, M_k]$ involving broken machines.
>
> Finally, the size-agnostic nature of our GNN-based policy network[7] is a crucial enabler for these dynamic scenarios. Changes in job or machine quantities do not require scheduling from scratch; MIStar can perform re-optimization from any valid existing solution [6].
>
> **Comparisons with dynamic scheduling algorithms**
>
> We appreciate the concern regarding the lack of comparisons with dynamic scheduling algorithms. To address this, we included results of two dynamic methods evaluated on static benchmarks. In future work, we consider to extend MIStar to dynamic scenarios and compare it with more dynamic approaches.
>
> **Table: Comparison with Dynamic Algorithms**
> ||la(edata)|la(vdata)|
> |-|-|-|
> |GASAVNS[10]|1143.78|-|
> |HRL[11]|-|954.15|
> |MIStar-400|1099.18|921.85|
> ## Reward curve analysis
> As we are unable to submit figures, we provide a descriptive analysis of the reward curve observed during training. The training process includes 20,000 episodes, which are grouped into batches of 20 to form 1,000 epochs. We then track the average return per epoch throughout training to assess the model’s learning progress over time.
>
> The curve shows frequent fluctuations in the initial phase (0–200 epochs), ranging approximately between 170 and 210. This high variability reflects the model's active exploration of the policy space. Since new training data is introduced every 20 epochs, the model must continuously adapt to different scheduling instances, which also contributes to these fluctuations. During the mid-phase (200–500 epochs), the curve becomes relatively more stable but still shows several noticeable drops in return, decreasing to around 150–180. This suggests potential local optima or instability in the training process. However, as the model gradually learns a more robust policy, it eventually escapes these suboptimal regions, with the reward converging to a stable range around 230 after approximately 600 epochs.
>
> ## Ablation on embedding architecture
> To accurately extract the key decision-making signals for the policy network—namely, the current scheduling state represented by operation and machine node embeddings—we construct a tailored representation module composed of multiple specialized graph encoders. We adopt GIN[8] to extract structural differences among operation nodes due to its strong discriminative power in graph structure learning. To model the distinct semantic constraints from neighboring nodes (i.e., job predecessors and machine predecessors), we employ a GAT[9] with $n_H$ attention heads, enabling the model to learn the varying importance of different types of neighbors[1][2]. Additionally, we use a HGAT[2] to learn machine embeddings that reflect machine workloads, which serve as critical heuristics for downstream policy decisions—such as guiding the agent to reassign operations from overloaded machines to less loaded ones to reduce makespan.
>
> We conducted ablation studies on the embedding architecture using four size instances from the SD2. Results below further confirm that each of these components is essential for learning informative and discriminative state representations that support effective policy learning.
> |||10×5||20×5||15×10||20×10||
> |-|-|-|-|-|-|-|-|-|-|
> |**Model**|**Iter.**|**Obj.**|**Time**|**Obj.**|**Time**|**Obj.**  |**Time**|**Obj.**|**Time**|
> |**w/o GIN**|100|348.31|4.97s|616.71|10.72s|491.78|18.12s|548.61|24.80s|
> ||200|347.06|10.78s|614.98|22.88s|484.36|37.58s|545.97|53.68s|
> ||400|345.73|20.56s|613.98|45.9s|477.8|1.45m|544.31|1.91m|
> |**w/o GAT**|100|350.86|5.82s|618.67|11.34s|492.76|17.85s|548.0|22.63s|
> ||200|349.89|9.46s|617.33|22.91s|486.74|36.97s|547.97|51.02s|
> ||400|348.33|21.41s|615.99|46.79s|480.66|1.29m|547.87|1.78m|
> |**w/o HGAT**|100|350.49|5.44s|618.15|11.39s|491.97|17.95s|535.78|23.89s|
> ||200|349.08|9.74s|616.56|23.01s|483.46|37.01s|532.83|52.47s|
> ||400|347.17|20.1s|614.97|46.38s|473.83|1.29m|529.92|1.67m|
> |**MIStar**|100|345.05|5.27s|615.12|11.55s|488.22|18.09s|530.93|25.76s|
> ||200|343.43|10.33s|613.75|23.50s|476.93|37.42s|528.92|53.25s|
> ||400|342.50|21.39s|612.73|47.93s|465.71|1.31m|525.49|1.88m|
> ## Robustness
> To tackle your robustness concern, we conducted 10 independent runs for each instance in Table 2, and reported the corresponding standard deviation to demonstrate robustness. Due to time constraints, we were unable to include similar statistical measures for the synthetic data in Table 1, but we will incorporate them in the final version to better illustrate performance stability.
> #### Table: Robust Performance across benchmarks（mean ± std）
> |             | mk          | la(rdata)   | la(edata)    | la(vdata)   |
> |-------------|-------------|-------------|--------------|-------------|
> | MISTar-100  | 178.36±0.49 | 963.35±0.92 | 1102.89±2.27 | 925.34±0.25|
> | MISTar-200  | 178.28±0.47 | 961.83±1.07 | 1102.78±2.23 | 924.4±0.22|
> | MISTar-400  | 178.08±0.41 | 960.3±0.98  | 1102.76±2.23 | 923.6±0.24|
> ##
> We are grateful to reviewer for their valuable suggestions. Combined with the discussions and new experimental evidence, these will guide us in revising the final version to boost its clarity and presentation effectiveness.
>
> [1]Zhang C, Cao Z, Song W, et al. Deep reinforcement learning guided improvement heuristic for job shop scheduling[J]. arXiv preprint arXiv:2211.10936, 2022.
> [2]Song W, Chen X, Li Q, et al. Flexible job-shop scheduling via graph neural network and deep reinforcement learning[J]. IEEE Transactions on Industrial Informatics, 2022, 19(2): 1600-1610.
> [3]Wang R, Wang G, Sun J, et al. Flexible job shop scheduling via dual attention network-based reinforcement learning[J]. IEEE Transactions on Neural Networks and Learning Systems, 2023, 35(3): 3091-3102.
> [4]Xie J, Gao L, Peng K, et al. Review on flexible job shop scheduling[J]. IET collaborative intelligent manufacturing, 2019, 1(3): 67-77.
> [5]An Y, Chen X, Gao K, et al. Multiobjective flexible job-shop rescheduling with new job insertion and machine preventive maintenance[J]. IEEE Transactions on Cybernetics, 2022, 53(5): 3101-3113.
> [6]Renke L, Piplani R, Toro C. A review of dynamic scheduling: context, techniques and prospects[J]. Implementing Industry 4.0: The Model Factory as the Key Enabler for the Future of Manufacturing, 2021: 229-258.
> [7]Wu Z, Pan S, Chen F, et al. A comprehensive survey on graph neural networks[J]. IEEE transactions on neural networks and learning systems, 2020, 32(1): 4-24.
> [8]Xu K, Hu W, Leskovec J, et al. How powerful are graph neural networks?[J]. arXiv preprint arXiv:1810.00826, 2018.
> [9]Veličković P, Cucurull G, Casanova A, et al. Graph attention networks[J]. arXiv preprint arXiv:1710.10903, 2017.
> [10]Fuladi S K, Kim C S. Dynamic events in the flexible job-shop scheduling problem: rescheduling with a hybrid metaheuristic algorithm[J]. Algorithms, 2024, 17(4): 142.
> [11]Lei, Kun, et al. "Large-scale dynamic scheduling for flexible job-shop with random arrivals of new jobs by hierarchical reinforcement learning." IEEE Transactions on Industrial Informatics 20.1 (2023): 1007-1018.

---

### Official Review · Reviewer_19Ys · 2025-07-03

**Clarity:** 2
**Significance:** 2
**Originality:** 3
**Rating:** 4
**Confidence:** 3

**Summary:**

This paper introduces MIStar, a Memory-enhanced Improvement Search framework designed to tackle the Flexible Job Shop Scheduling Problem (FJSP). Traditional deep reinforcement learning approaches often fall short of optimal solutions for FJSP, while improvement-based methods, though effective, face challenges with state representation and efficient search strategies due to flexible machine allocation. MIStar addresses these issues by employing a novel heterogeneous disjunctive graph for accurate solution representation, a memory-enhanced heterogeneous graph neural network (MHGNN) for improved decision-making through historical data, and a parallel greedy search strategy for efficient exploration.

**Questions:**

1. What is the run time of OR-Tools in Table 1?
2. How do you plan to incorporate additional constraints (e.g., machine breakdowns) into your model?
3. What is the actual training time for MIStar? Do you need to retrain the model for different benchmarks? How well can the model transfer knowledge from one benchmark to another?
4. Given that MIStar also requires an initial solution, how does the quality of this initial solution affect MIStar's overall performance? Do you have any additional experimental results to illustrate this impact?
5. Do you have scalability results for larger benchmarks where even OR-Tools might struggle to generate good solutions within a reasonable timeframe?

**Ethical Concerns:**

["NO or VERY MINOR ethics concerns only"]

**Final Justification:**

The authors have addressed most of my concerns, so I would like to raise my score.

**Limitations:**

See above.

**Paper Formatting Concerns:**

- There should be colons after the table and figure caption numbers.

**Quality:**

3

**Strengths And Weaknesses:**

- Strength
  - A DRL-based improvement heuristic framework specifically designed for the FJSP problem.
- Weakness
  - The benchmarking examples seem pretty small (40x10). It is challenging to ascertain the model's true scalability given that even conventional optimization tools like OR-Tools can achieve optimal solutions in such settings. For a more comprehensive evaluation, the authors should consider larger, practical scheduling problems often found in domains like systems and networking, which involve datasets with thousands or millions of nodes.
  - Real-world FJSP often involves multiple objectives (e.g., minimizing total flow time, machine idle time) and additional constraints (e.g., machine breakdowns, dynamic job arrivals). The current MDP formulation, reward function design (which explicitly focuses on makespan improvement and visit penalties), and the Nopt2 neighborhood structure are tailored to makespan minimization. The adaptability of MIStar to these more complex and constrained scenarios is not explored.
  - The paper does not delve into the impact of different initial solution generation methods on MIStar's performance or how robust the framework is to poor initial solutions.

---

> ### Author Rebuttal · Authors · 2025-07-31
>
> We appreciate the reviewers' feedback and address each comment below:
> ## Regarding performance, benchmark size, and scalability
> We thank the reviewer for these important questions. Our choice of benchmarks, including instances up to 40x10, was intended to align with standard yet challenging problems in the FJSP literature[2][3].We respectfully note that due to the problem's NP-hard nature, these instances are far from trivial.
>
> To the reviewer's point about OR-Tools, its runtime was limited to 30 minutes for experiments in Table 1. The results already show that OR-Tools' ability to find and prove optimality degrades significantly as instance size increases; Due to the complexity of FJSP, even on small-to-medium instances (10×5, 20×5, 15×10, 20×10), OR-Tools fails to find optimal solutions within a reasonable time for most cases. For example, the *optimality rate* on 20×5, 30×10, and 40×10 instances is 0%. This indicates these are precisely the sizes where powerful learned heuristics become critical.
>
> To more directly address the scalability concern, we conducted new experiments on even larger instances (up to 1,500 operations). For each problem size, 10 instances were randomly generated, and both MIStar and OR-Tools were given a one-hour time limit per instance.
>
> The results, summarized below, highlight two critical advantages of our approach. First and most strikingly, on the highly complex 50x30 instances, **OR-Tools failed to produce any feasible solution within the time limit, whereas MIStar consistently found high-quality solutions**. This demonstrates MIStar's superior robustness and scalability on truly challenging problems.
>
> Second, for the other large instances where OR-Tools did find a solution, **MIStar achieves comparable solution quality in a small fraction of the time**. For example, on the 100x10 instances, MIStar reaches a solution with a 66.03% optimality gap in just under 16 minutes, while OR-Tools requires a full hour to achieve a 62.22% gap. This massive speed-up, combined with the strong generalization from a model trained only on smaller 20x10 instances, confirms that MIStar provides an effective and highly scalable solution for large-scale FJSP.
> #### Table: Generalization Performance on Larger-Scale Instances
> |Instance Size|Method|Avg. Objective|Avg. Time (min)|Optimality Gap (%)|
> |:-|:-|:-:|:-:|:-:|
> |50×15|OR-Tools|881.45|60.0|39.05%|
> |(LB:537.2)|MIStar-200|969.7|10.6|44.60%|
> |60×15|OR-Tools|1075.1|60.0|46.94%|
> |(LB:570.2)|MIStar-200|1109.8|14.2|48.62%|
> |100×10|OR-Tools|1922.65|60.0|62.22%|
> |(LB:726.5)|MIStar-200|2138.4|15.9|66.03%|
> |50×30|OR-Tools|Infeasible|60.0|-|
> |(LB:N/A)|MIStar-200|1040.1|60.0|-|
> ## Adaptation to additional constraints
> We thank the reviewer for this important point. We would first like to clarify that our focus on the standard, makespan-oriented FJSP is a deliberate methodological choice. It aligns with the mainstream research paradigm in learning-based scheduling[1,2,3], which prioritizes creating powerful solvers for the core NP-hard problem as a foundational step. Even the static, single-objective FJSP remains a formidable challenge, and developing effective heuristics for it is an active and critical area of research [1,4].
>
> That being said, we fully agree that adaptability to real-world dynamics is crucial. Our core components are highly modular and allow for the incorporation of additional constraints and objectives with minimal, targeted modifications. We are happy to illustrate this adaptability with two representative examples:
> #### **Handling machine breakdowns：**
> Our framework can elegantly manage unexpected events like machine failures. This is possible because our state representation and action space are inherently flexible.
> * **State:** The machine status is a feature in our graph representation, allowing the breakdown to be modeled by updating the corresponding node. The agent, trained to minimize makespan, will naturally avoid assigning operations to unavailable machines or those with infinite processing times, highlighting the effectiveness of our rich, feature-based state representation.
> * **Action:** All operations currently assigned to failed machine must be re-assigned. Broken machines are masked out when generating candidate actions based on the Nopt2 neighborhood structure. As a result, invalid actions $[O_m, O_n, M_k]$ involving broken machines in the third dimension are effectively filtered out. This ensures the agent always operates within valid constraints.
> * **Reward:** To manage the trade-off between efficiency and stability during rescheduling, the reward function can be augmented. A stability term[5] can be incorporated into the reward to minimize the differences in operation start times between the original and revised schedules $\sum_{j=1}^{|J|} \sum_{i=1}^{n_j} \left| st_t - st_{t-1} \right|$, thus encouraging minimal disruption to the former schedule[5,6].
> #### **Handling dynamic job arrivals：**
> MIStar is exceptionally well-suited for dynamic re-optimization, a key advantage of our improvement-based approach over constructive methods.
> * **State:** The flexibility of our graph structure allows new jobs to be seamlessly integrated by expanding the state matrices to include their operations and associated constraints. To maintain focus and efficiency, completed operations can be pruned from the state, ensuring the agent always reasons over the most relevant part of the problem.
> * **Action:** Time-based constraints are easily handled. To respect arrival times, we simply filter out any action that would schedule a job's first operation before it is ready. The problem's inherent precedence constraints, already encoded in our graph, automatically handle the sequencing for all subsequent operations.
> * **Reward:** A penalty term can be added to minimize earliness and tardiness across jobs[7], in order to ensure on-time delivery. Additionally, the robustness term above encourages minimal disruption to the earlier schedule[5].
>
> Finally, the size-agnostic nature of our GNN-based policy network [9] is a crucial enabler for these dynamic scenarios. Changes in job or machine quantities do not require scheduling from scratch; MIStar can perform re-optimization from any valid existing solution [8]. Furthermore, our memory mechanism remains robust, as the incidence matrices from schedules of different sizes can be aligned via simple padding or cropping before similarity comparison. This architectural foresight ensures MIStar is not just a static solver, but a truly adaptable scheduling framework.
> ## Training cost and generalization
> MIStar is trained on synthetic data for 20,000 episodes (batch size 20, 100 steps/episode), which takes approximately 4 to 13 days depending on the instance size. A key strength of our approach is that the model **does not need to be retrained for different benchmarks**. As demonstrated in Tables 1 and 2, our single trained model demonstrates strong scalability and adaptability across benchmarks (Table 2), achieving the highest average performance across instances and generalizing well to larger, unseen instances (30×10 and 40×10 in Table 1).
> ## Regarding the impact of the initial solution
> To address the reviewer's question, we have performed a systematic evaluation of MIStar's performance starting from initial solutions of varying quality. This new experiment, detailed in Appendix H (Table 4), uses six different initialization strategies, from DRL-based methods to simple priority rules and random solutions.
>
> The results show that while better initial solutions can lead to higher-quality final schedules within a fixed iteration count, MIStar consistently yields significant improvements regardless of the starting point. Notably, it achieves larger relative improvements when starting from poorer solutions. This demonstrates the robustness of our learned heuristic and its ability to effectively navigate the solution space from diverse initial conditions.
> ##
> We sincerely appreciate the reviewer’s insightful comments in the rebuttal process. Incorporating these insights, along with the new experimental data and discussions, we will revise the final version of the paper to enhance its clarity and strengthen its overall presentation.
>
> [1]Deep reinforcement learning guided improvement heuristic for job shop scheduling, 2022.
> [2]Flexible job-shop scheduling via graph neural network and deep reinforcement learning, 2022.
> [3]Flexible job shop scheduling via dual attention network-based reinforcement learning, 2023.
> [4]Review on flexible job shop scheduling, 2019.
> [5]Dynamic scheduling in flexible job shop systems by considering simultaneously efficiency and stability, 2010.
> [6]Robust scheduling for multi-objective flexible job-shop problems with random machine breakdowns, 2013.
> [7]Multiobjective flexible job-shop rescheduling with new job insertion and machine preventive maintenance, 2022.
> [8]A review of dynamic scheduling: context, techniques and prospects, 2021.
> [9] A comprehensive survey on graph neural networks, 2020.

---

> ### Author Response · Authors · 2025-08-07
>
> I hope this message finds you well. As the discussion period is nearing its end with **less than three days remaining**, I want to ensure we have addressed all your concerns satisfactorly. If there are any additional points or feedback you'd like us to consider, please let us know. Your insights are invaluable to us, and we're eager to address any remaining issues to improve our work.
>
> Thank you for your time and effort in reviewing our paper.

---

> > ### Comment · Reviewer_19Ys · 2025-08-07
> >
> > I appreciate the authors' detailed response. The new experiments and clarifications have answered most of my questions. Please also consider incorporating them in your next paper draft.

---

> > > ### Author Response · Authors · 2025-08-08
> > >
> > > We sincerely appreciate the reviewer for acknowledging our response, and we will incorporate them in the revised paper. We thank you again for raising the score and supporting our work!

---

### Decision · Program_Chairs · 2025-09-17

**Decision:**

Accept (poster)

**Comment:**

This paper introduces MIStar, a Memory-enhanced Improvement Search framework designed to tackle the Flexible Job Shop Scheduling Problem (FJSP). Based on comments from reviewers, the merits of this paper include: (1) The integration of memory mechanisms with heterogeneous graph neural networks is innovative and well-suited for FJSP. Particularly, bring memory-enhanced GNNs into DRL-based improvement heuristics for FJSP, addressing challenges that existing methods often overlook.(2) The proposed method outperforms the traditional heuristics and DRL-based constructive methods across both synthetic and real-world benchmarks. (3) Each component of the proposed MIStar framework is well-motivated by a specific challenge in applying improvement methods to FJSP.

One reviewer raised a concern regarding the practical applicability of the approach in industrial settings. However, this concern is relatively minor given that FJSP is a widely recognized and longstanding benchmark in job shop scheduling research.

Following the rebuttal, the paper received one Accept and no negative scores. Given the strengths in both methodological innovation and empirical results, I recommend acceptance of this paper.